# LLM-Guided Self-Supervised Tabular Learning With Task-Specific Pre-text Tasks

**Sungwon Han**                                            *lion4152@gmail.com*
*Korea Advanced Institute of Science and Technology (KAIST)*

**Seungeon Lee**                                           *marinearchon159@gmail.com*
*Korea Advanced Institute of Science and Technology (KAIST)*

**Meeyoung Cha**                                           *mia.cha@mpi-sp.org*
*Max Planck Institute for Security and Privacy*

**Sercan Ö. Arik**                                         *soarik@google.com*
*Google Cloud AI*

**Jinsung Yoon**                                           *jinsungyoon@google.com*
*Google Cloud AI*

**Reviewed on OpenReview:** *https://openreview.net/forum?id=jXcx2oAIbw*

## Abstract

One of the most common approaches for self-supervised representation learning is defining pre-text tasks to learn data representations. Existing works determine pre-text tasks in a "task-agnostic" way, without considering the forthcoming downstream tasks. This offers an advantage of broad applicability across tasks, but can also lead to a mismatch between task objectives, potentially degrading performance on downstream tasks. In this paper, we introduce TST-LLM (Task-specific Self-supervised Tabular learning with LLMs), a framework that effectively reduces this mismatch when the natural language-based description of the downstream task is given without any ground-truth labels. TST-LLM instructs the LLM to use the downstream task's description and meta-information of data to discover features relevant to the target task. These discovered features are then treated as ground-truth labels to define "target-specific" pre-text tasks. TST-LLM consistently outperforms contemporary baselines with win ratios of 95% and 81%, when applied to 22 benchmark tabular datasets, including binary and multi-class classification, and regression tasks.

## 1 Introduction

Obtaining unlabeled data for machine learning is typically more scalable and cheaper than gathering labeled data in real-world applications. Self-supervised representation learning, which was proposed to extract useful information from unlabeled data, enhances the performance of downstream tasks by obtaining superior representations (Chen et al., 2020; Oord et al., 2018; Tschannen et al., 2019). A common approach for self-supervised representation learning is utilizing a pre-text task based on the type or characteristics of the data, and then learning representations by optimizing the objective of the pre-text task (instead of the downstream task) (Assran et al., 2022; Kim et al., 2018; Zhang et al., 2017). For example, in computer vision domain, such a pre-text task can be defined to estimate the degree of rotation applied to an image (Gidaris et al., 2018). Other works have defined augmentations that do not deform the contents of images, such as horizontal flips or color jittering, to learn representations that are invariant to these modifications (Grill et al., 2020; Han et al., 2020; Zbontar et al., 2021).

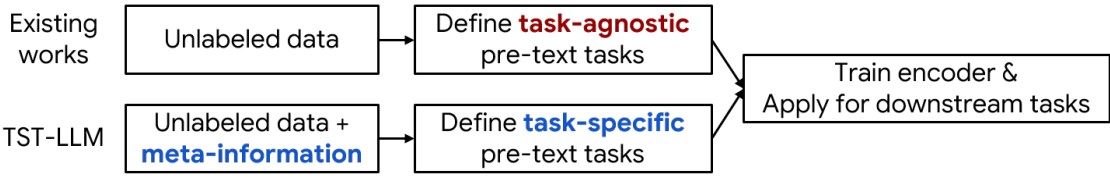

Figure 1: Illustration comparing existing self-supervised representation learning methods with our approach.

These pre-text task-based methods have also been extended to tabular data – specifically, efforts have been made to adapt pre-text tasks that were previously limited to images and text. Common tasks for tabular data include corrupting or masking data and then reconstructing the original sample from the representation (Yoon et al., 2020; Wu et al., 2024), or designing augmentations suited to tabular data to perform contrastive learning tasks (Bahri et al., 2022; Somepalli et al., 2022). The inductive bias provided by the pre-text tasks plays a role in preemptively removing spurious correlations or noisy information within tabular data.

Despite the success of using pre-text tasks in tabular learning, a fundamental limitation remains in the task mismatch between the pre-text tasks optimized by representation learning and the actual downstream tasks (Sui et al., 2024). This is because the pre-text tasks in representation learning are determined in a "task-agnostic" way that are oblivious to the forthcoming downstream tasks (see the upper part of Figure 1). Task-agnostic definitions offer the advantage of broad applicability across tasks, but also potentially risks including noisy irrelevant information or eliminating critical information for the downstream task. For example, in the diabetes classification task (Smith et al., 1988), height and weight were important correlated features for classification. However, applying augmentations like randomly masking these features in tabular data with high variable correlation can create unrealistic samples and result in the loss of crucial correlation information (Sui et al., 2024; Hajiramezanali et al., 2022). Such fallacies impair the performance of the downstream task.

Our study seeks to address the issue of the task-objective mismatch by defining pre-text tasks in a *task-specific* rather than task-agnostic manner using the natural-language based description of the downstream task. The description includes the task objective (e.g., "Does this patient have diabetes? Yes or no?") and the answer candidates (e.g., "Yes" or "No"). Using this information, we propose, TST-LLM (Task-specific Self-supervised Tabular learning with LLMs), which aims to improve representation learning via LLM-discovered features without incorporating any ground-truth labels or label statistics (see the lower part of Figure 1). By leveraging the prior knowledge of the LLM, we explore the relationship between the task and data features from their natural language-based descriptions. This process aims to determine which combinations or transformations of original data features can yield meaningful information for solving the task. Then, the new features created through the LLM's prior knowledge are used to generate the ground-truth labels for the pre-text tasks that train the representation. For example, in the task of predicting whether a patient has diabetes, newly discovered features such as "`bmi = weight / height`", based on prior knowledge, are likely to have a higher correlation with the label. Learning with these features provides additional task-relevant information, such as the importance of original features to the task and the correlation between them. Further, this enables learning representations that are better aligned with the target downstream task, as opposed to conventional pre-text tasks that are defined solely based on the data structure without considering the underlying problem.

TST-LLM consists of two main stages. In the first stage, target task's textual description, meta-information of data (e.g., feature names and descriptions), and text-serialized unlabeled data are used to construct prompts. Then, they are fed into an LLM to extract new task-relevant features. This process is repeated, while previously extracted features are excluded at the next iteration to ensure the diversity of feature synthesis. In the second stage, the discovered features are considered as ground-truth labels to define a pre-text task. We use supervised contrastive learning (Khosla et al., 2020) to perform multi-task learning for each label, learning useful representations. Additionally, we introduce a process for selecting a diverse feature set from the discovered features that are distinctly aligned with both the original data and each other for computational efficiency.

A key advantage of TST-LLM is its simplicity; it can be applied to any problem as long as a task description and feature descriptions are defined in natural language. We demonstrate that the features discovered

by the LLM are meaningful and relevant to the actual target task. Our model consistently outperforms contemporary baselines with win ratios of 95% and 81%, when applied to 22 benchmark tabular datasets including binary and multi-class classification, and regression tasks.

The remainder of the paper is structured as follows: Section 2 reviews related work, Section 3 presents our method, Section 4 discusses experimental results, and Section 5 concludes with key findings and future directions. Our code is available on Github[1].

## 2 Related Work

**Self-supervised representation learning for tabular data.** New advancements in self-supervised representation learning enable the discovery of meaningful representations from unlabeled data across wide modalities, from images (Caron et al., 2020; Wen et al., 2022; Wu et al., 2018) to texts (Gao et al., 2021; Kenton & Toutanova, 2019; Radford et al., 2019), audio (Mittal et al., 2022; Owens & Efros, 2018), and most recently to tabular data (Balestriero et al., 2023; Gharibshah & Zhu, 2022). One of the common ways of self-supervised learning is to define a pre-text task on an unlabeled dataset to facilitate learning. According to the literature (Gharibshah & Zhu, 2022), pre-text tasks for tabular data can be broadly classified into three types. The first category, invariance learning, involves defining a positive view of a given sample and learning the representation invariance between them. The positive view of the sample can be created using weak augmentations that do not distort the original content (Bahri et al., 2022; Somepalli et al., 2022) or by selecting samples with similar characteristics from the training data (Nam et al., 2023b). The second category, predictive learning, includes methodologies that generate explicit labels from the dataset and train the model to predict these labels. For example, masking or corrupting data and then using the original data for reconstruction as a label (Wu et al., 2024; Yoon et al., 2020). Some studies also proposed pre-text tasks on various publicly available benchmark datasets or synthetic datasets and then performing transfer learning for downstream tasks (Hollmann et al., 2023a; Wang & Sun, 2022). The last category includes a hybrid approach combining invariance and predictive learning (Ucar et al., 2021; Zhu et al., 2023).

All of the above methods define pre-text tasks in a task-agnostic manner, which can lead to inconsistencies with the actual objectives of the downstream tasks that can ultimately hinder performance. Our study proposes a method to effectively reduce this inconsistency by generating task-specific pre-text tasks using the description of the downstream task.

**Tabular learning with large language model.** LLMs can be applied to various domains by leveraging their prior knowledge and generalizability to handle unseen tasks (Anil et al., 2023; OpenAI, 2023). Recent research has explored serializing tabular data into natural language-based text to solve tabular tasks (Dinh et al., 2022; Hegselmann et al., 2023; Wang et al., 2023). Task description and meta-information of the tabular data, such as feature names and descriptions, guide the LLM on which features to focus on in order to solve the problem. Especially, the rich prior knowledge of LLMs boosts performance in few-shot settings, where labeled data is scarce (Hegselmann et al., 2023). Previous works (Dinh et al., 2022; Hegselmann et al., 2023; Nam et al., 2023a) broadly cover the case of in-context learning while calibrating the prompts without further training of the LLM, and the case of applying parameter-efficient fine-tuning techniques like $(IA)^3$ (Liu et al., 2022) or LoRA (Hu et al., 2021).

Our method leverages the prior knowledge and reasoning ability of LLMs to aim for task-specific pre-text task generation through the downstream task description and the meta-information of data. We only use prompt engineering, enabling operation with limited access to LLMs, similar to an API.

## 3 Method

**Problem formulation.** Let's consider a tabular dataset with $d$-dimensional input features $\mathcal{D} = \{\mathbf{x}_i\}_{i=1}^N$, where $\mathbf{x}_i \in \mathbb{R}^d$ and no ground-truth labels are provided. A downstream task description in natural language, $E_{\text{task}}$, and the names and short descriptions of each feature, $E_{\text{name}} = \{e_{\text{name}}^j\}_{j=1}^d$ and $E_{\text{desc}} = \{e_{\text{desc}}^j\}_{j=1}^d$, are provided. The model aims to train an encoder $f$ that maps raw input $\mathbf{x}_i$ into a informative data

---
[1]https://github.com/Sungwon-Han/TST-LLM

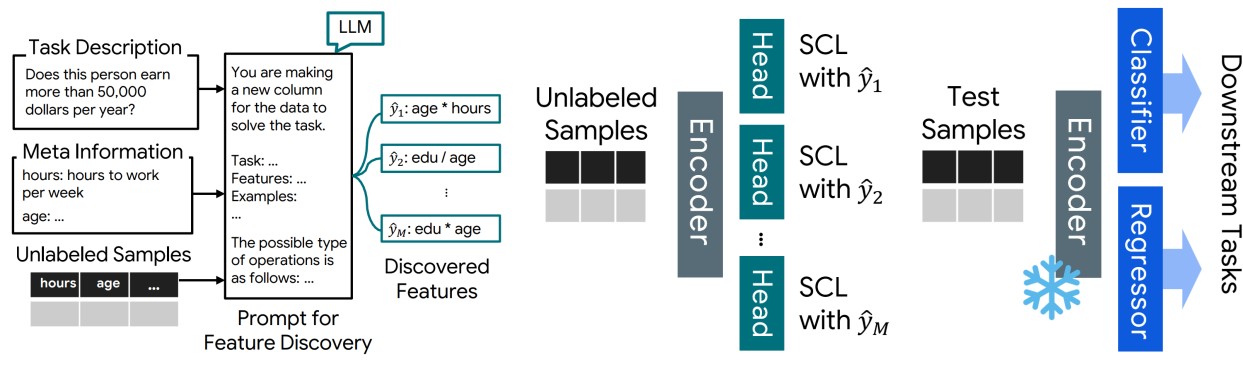

(a) LLM-guided feature discovery     (b) Task-specific learning     (c) Apply for downstream tasks

Figure 2: Illustration of TST-LLM. (a) It utilizes the downstream task description and the meta-information of the data to discover features relevant to the task. (b) Subsequently, the discovered features are treated as ground-truth labels to perform task-specific representation learning. (c) The trained encoder is then frozen and used to solve downstream tasks by attaching a simple classifier/regressor.

representations $f(\mathbf{x}_i)$. The trained encoder $f$ is then frozen and used to solve the downstream task by attaching a simple classifier/regressor on top of $f(\mathbf{x}_i)$. The downstream tasks can be binary or multi-class classification and regression.

Figure 2 and Algorithm 1 illustrate our model. TST-LLM takes in downstream task description and meta information to define a pre-text task that aligns with the downstream task objectives, and performs representation learning via this task. Initially, the model passes the task description $E_{\text{task}}$ along with meta-information of data $E_{\text{name}}$ and $E_{\text{desc}}$ to the LLM to generate potentially relevant features through combinations or transformations of original features (Section 3.1). These generated features are set as target labels for the pre-text task, which are then used to train the encoder through multi-task contrastive learning (Section 3.2). Details of each stage are described below.

---

**Algorithm 1:** Algorithm for TST-LLM.

**Input** : Original dataset $\mathcal{D}$, Large language model backbone $LLM$, Encoder $f$, Original input feature set $\mathcal{Y}$, Number of features to select $M$, Entropy threshold $t_{\text{ent}}$, Meta-information $E_{\text{task}}$, $E_{\text{name}}$, and $E_{\text{desc}}$.

**Output** : Trained encoder $f$.

1   $\hat{\mathcal{Y}}_{init} = \text{FeatureDiscovery}(LLM, \mathcal{Y}, E_{\text{task}}, E_{\text{name}}, E_{\text{desc}})$ ;      `// LLM-guided feature discovery`
2   $\hat{\mathcal{Y}} = \text{FeatureSelection}(\hat{\mathcal{Y}}_{init}, \mathcal{D}, M, t_{\text{ent}})$ ;      `// Feature selection with minimum redundancy`
3   Define a set of projection heads $\mathcal{G} = \{g^1, g^2, \dots, g^M\}$
4   Optimize $f, \mathcal{G}$ using $\mathcal{L}_{\text{SCL-multi}}$ (Eq. 3) ;      `// Representation learning with discovered features`
5   Freeze $f$ for downstream tasks

---

## 3.1   LLM-Guided Feature Discovery with Task Description

The model generates data-related features from task description and meta information utilizing the prior knowledge and reasoning abilities of an LLM. Feature discovery process involves running multiple LLM inferences. The designed prompt consists of four main components: data description summary, operation instruction, diversity enforcement, and response instruction (see Figure 3 and Appendix A.2 for the example prompt).

**Data description summary.** This component provides a basic data description for feature discovery (see blue part in Figure 3). It includes the downstream task's description $E_{\text{task}}$ (e.g., "Does this person earn more than 50,000 dollars per year? Yes or no?") as well as feature names and descriptions $E_{\text{name}}$, $E_{\text{desc}}$ (e.g., "hours-per-week": "the hours an individual has reported to work per week"). Similar to other works (Hegselmann et al., 2023), we serialize the sample data as in-context demonstration, giving hints on

> You are a data engineer. Given the task description and the list of features and data examples, you are making a new column for the data which is informative to solve the task.
>
> Task: [Downstream Task Description]
> Features: [Feature Descriptions]
> Examples: [Serialized Examples]
>
> Given a type of operations below, generate 5 new columns which are the most informative to solve the task using operations. Refer to the examples when generating features. Only use features listed in the feature description. Note that multiple operations can be nested to generate a new column.
>
> The possible type of operations is as follows:.
> [Operation Descriptions]
>
> You also have some new example features generated with these modules.
> Example Features:
> Index | Feature_name | Feature_desc
> [Features From Previous Trial]
>
> You must write new feature that is different from all above examples features with respect to both names and descriptions.
>
> Format of response for 5 new columns:
> ——
> Thought 1: Any reasons why the following new feature would be helpful for the task
> New feature 1: Type of operation | New_column_name | One line pseudo code for generating columns
> ...
> Thought 5: ...
> New feature 5: ...
> ——

Figure 3: Prompt for feature discovery. Text in blue corresponds to data description summary part, red text to operation instruction, teal text to diversity enforcement, and brown text to response instruction part.

the scale and format of the data. Given the data $\mathbf{x}$, serialization is applied as:

$$\text{Serialize}(\mathbf{x}, E_{\text{name}}) = \text{`` } e^1_{\text{name}} \text{ is } \mathbf{x}^1. \ \cdots \ e^d_{\text{name}} \text{ is } \mathbf{x}^d.\text{''}, \tag{1}$$

where the superscript represents the vector's index value.

**Operation instruction.** This component guides the LLM on possible operations for feature discovery (see red part in Figure 3). It encourages the LLM to search only for feasibly-generated features, preventing erroneous behaviors (e.g., generating features that cannot be created from original data features or establishing ambiguous feature definitions). The operations used are as follows:

- Transformations: Transform the feature value with one of the following operators: absolute, logarithm, square root, sigmoid, or frequency.
- Numerical Operations: Conduct arithmetic operations from multiple numerical features.
- Categorical Operations: Combine two categorical features to generate a new feature.
- Mixed-type Operations: Combine categorical and numerical features to generate a new one. For example, the model can discretize a numerical feature into a categorical one, allowing for categorical operations between the two features.

**Diversity enforcement.** Instead of concluding feature discovery with a single query, we aggregate features from multiple queries to find useful features. However, we want to avoid the model from discovering duplicate features over multiple trials. To ensure diverse search, we provide additional instructions to prevent the LLM from selecting features identified in previous attempts (see teal part in Figure 3). We include descriptions via one-line pseudo-code, along with feature names, to prevent the LLM from simply renaming and selecting the same features. This component is integrated in all iterations except for the initial iteration.

**Response instruction.** This component includes instructions on how the LLM should format its response (see brown part in Figure 3). The format includes the type of operation, the feature's name, and a one-line pseudo-code necessary to regenerate the feature (e.g., "Numerical Operations | capital_diff | Subtract capital-loss from capital-gain to get the net capital difference"). Setting the response format facilitates easier parsing later on and also gives further evidence on each feature discovery, explaining why the particular feature was selected upon response.

The output text from the LLM prompt is parsed to generate the discovered feature. The generation process is automatically carried out using the LLM, which uses one-line pseudo-code and data to generate function code for producing the feature. The prompt used for automated generation is in the Appendix A.4.

## 3.2 Representation Learning with Discovered Features

The discovered features are semantically related to the target downstream task based on LLM's prior knowledge. We considered these features as ground-truth labels $\hat{y}$ to define a pre-text task aligned with the target downstream task. Although TST-LLM is agnostic to the choice of learning methods[2], we adopt supervised contrastive learning for its generalizability to downstream tasks (Graf et al., 2021; Khosla et al., 2020). We define the projected representation of sample $\mathbf{x}_i$ as $\mathbf{z}_i = g(f(\mathbf{x}_i))$, where $f$ is the encoder and $g$ is the projection head. According to the literature (Khosla et al., 2020), given a batched set of $N_b$ samples with a pseudo label $\mathcal{B} = \{\mathbf{x}_i, \hat{y}_i\}_{i=1}^{N_b}$, the supervised contrastive loss with a temperature $\tau$ is defined as below. For numerical features among the discovered features, we transformed them into discrete features using 1-dimensional $k$-means clustering with $k = 10$.

$$\mathcal{L}_{\text{SCL}} = -\frac{1}{|\mathcal{B}|} \sum_{i \in \mathcal{B}} \sum_{j \in \mathcal{B}, j \neq i} \mathbf{1}_{\hat{y}_i = \hat{y}_j} \log \frac{\exp(\mathbf{z}_i \cdot \mathbf{z}_j / \tau)}{\sum_{k \in \mathcal{B}} \mathbf{1}_{i \neq k} \exp(\mathbf{z}_i \cdot \mathbf{z}_k / \tau)} \tag{2}$$

In our model, multiple features are discovered, and correspondingly, there are multiple labels available for supervised contrastive learning. We utilize a multi-task learning approach to train the encoder, by defining a projection head for each ground-truth label and simultaneously performing supervised contrastive learning for each label. Specifically, given a set of $M$ discovered features, $\hat{\mathcal{Y}} = \{\hat{y}^1, \hat{y}^2, \ldots, \hat{y}^M\}$, we define a set of projection heads $\mathcal{G} = \{g^1, g^2, \ldots, g^M\}$. Subsequently, the encoder $f$ optimizes the following loss:

$$\mathcal{L}_{\text{SCL-multi}} = \frac{1}{|M|} \sum_{m=1}^{M} \mathcal{L}_{\text{SCL}}^m, \tag{3}$$

where each $\mathcal{L}_{\text{SCL}}^m$ is a supervised contrastive loss computed with the respective projection head $g^m$ for the corresponding label set $\hat{y}^m$.

**Feature selection with minimum redundancy.** Multi-task learning on all features generated by the LLM can be computationally heavy. Not all features are informative, and those that closely correlate with original features tend to lose their value as pre-texts. Furthermore, high correlation among the generated features could diminish the benefits of multi-task learning. To address this, we eliminate features that do not contribute meaningful information and carefully choose a diverse set of features with minimal redundancy, thereby reducing computation costs (see Algorithm 2).

First, we define *uninformative* features as those with the lowest entropy values in the distribution. For example, if a feature is predominantly assigned to only one class across all samples (i.e., low entropy), the amount of information that can be learned from this feature is also limited. After calculating the entropy of each discovered feature, those with an entropy below a specific threshold (i.e., $t_{\text{ent}}$) are eliminated. The filtering threshold $t_{\text{ent}}$ is set to 0.7, taking into account the entropy distribution of the entire feature set.

For the remaining features, the model selects a feature set that minimizes redundancy. The initial choice is the feature with the smallest correlation to the original data. The remaining features are then selected among the ones with the smallest correlation with the original data, including the previously added features. This

---

[2]See the Appendix E for the comparison with alternative learning methods (including reconstruction).

process is repeated until a predetermined number of features, $M$, are selected. Cramer's V value (Cramér, 1999) is used to measure the correlation after discretizing all numerical features in the same manner as $\hat{y}$. This approach ensures that the selected features have low correlation with the original data while maintaining diversity among the discovered features, enabling efficient multi-task learning (see Table 1 for an analysis of feature diversity). Refer to the Appendix J for example features generated and selected by our method.

---

**Algorithm 2:** Algorithm for feature selection with minimum redundancy.

**Input** : Initial feature set $\hat{\mathcal{Y}}_{\text{init}}$, Number of features to select $M$, Original dataset $\mathcal{D}$, Entropy threshold $t_{\text{ent}}$.
**Output** : Selected feature set $\hat{\mathcal{Y}}$

**1** $\hat{\mathcal{Y}} \leftarrow \emptyset$
**2** $\hat{\mathcal{Y}}_{\text{filtered}} \leftarrow \{\hat{y} \mid \text{Entropy}(\hat{y}) \geq t_{\text{ent}},\ \hat{y} \in \hat{\mathcal{Y}}_{\text{init}}\}$ ;  // Filter low entropy features
**3** **while** $|\hat{\mathcal{Y}}| < M$ **do**
**4** $\quad$ $\Phi \leftarrow \emptyset$
**5** $\quad$ **for** $\hat{y} \in \hat{\mathcal{Y}}_{filtered}$ **do**
**6** $\quad\quad$ $\phi_y \leftarrow \max(\text{CramersV}(\mathcal{D}, \hat{y}))$ ;  // Compute redundancy of the feature
**7** $\quad\quad$ $\Phi \leftarrow \Phi \cup \{(\phi_y, \hat{y})\}$
**8** $\quad$ **end**
$\quad$ /* Select features with minimum redundancy */
**9** $\quad$ $\hat{\mathcal{Y}}_{\text{selected}} \leftarrow \{\hat{y} \mid \phi_y = \min_{\phi_y}(\Phi), (\phi_y, \hat{y}) \in \Phi\}$
**10** $\quad$ $\hat{\mathcal{Y}}_{\text{filtered}} \leftarrow \hat{\mathcal{Y}}_{\text{filtered}} - \hat{\mathcal{Y}}_{\text{selected}}$
**11** $\quad$ $\mathcal{D} \leftarrow \mathcal{D} \cup \hat{\mathcal{Y}}_{\text{selected}}$
**12** $\quad$ $\hat{\mathcal{Y}} \leftarrow \hat{\mathcal{Y}} \cup \hat{\mathcal{Y}}_{\text{selected}}$
**13** **end**

---

## 4 Experiment

We evaluate `TST-LLM` across multiple tabular datasets with various downstream tasks. Through our experiments, we discuss which components of the model contributed to performance enhancements and how our model operates. Due to space constraints, full results, computational cost analysis, results with alternative learning objectives, and other extra analyses can be found in the Appendix.

### 4.1 Performance Evaluation

**Datasets.** Our study used a diverse range of downstream tasks in terms of size and complexity. Since our model requires task and feature descriptions, we selected a total of 22 benchmark datasets along with appropriate meta-information, referring to previous works – FeatLLM (Han et al., 2024) and TabLLM (Hegselmann et al., 2023): Adult (Asuncion & Newman, 2007), Balance-scale (Siegler, 1994), Bank (Moro et al., 2014), Blood (Yeh et al., 2009), Car (Kadra et al., 2021), Communities (Redmond, 2009), Credit-g (Kadra et al., 2021), Diabetes (Smith et al., 1988), Eucalyptus (Bulloch et al., 1991), Forest-fires (Cortez & Morais, 2008), Heart (fedesoriano, 2021), Junglechess (van Rijn & Vis, 2014), Myocardial (Golovenkin et al., 2020), Tic-tac-toe (Aha, 1991), Vehicle (Mowforth & Shepherd), Bike (Fanaee-T, 2013), Crab (Sidhu, 2021), Housing (Pace & Barry, 1997), Insurance (Datta, 2020), Wine (Cortez & Reis, 2009), Sequence-type, and Solution-mix. Descriptive statistics and task descriptions for each dataset are available in the Appendix A.1 and B. Among them, 15 datasets were used for classification problems and 7 for regression problems; Two datasets—Sequence-type and Solution-mix—are synthetic, ensuring they are not included in the LLM's pre-training corpus. Evaluation results on additional datasets are included in Appendix F.1.

**Baselines.** Our model was compared to nine baselines, all of which were trained under the same unsupervised setting as our experimental setup. (1) Raw Data: Uses the data as-is for the downstream task without any representation learning; (2) AutoEncoder (Baldi, 2012): Utilizes a pre-text objective that projects data into embeddings and reconstructs the original data; (3) SimSiam (Chen & He, 2021): Trains to minimize the embedding distance between a sample and its augmented version using a siamese network structure; (4) SCARF (Bahri et al., 2022): Employs self-supervised contrastive learning to train augmentation-invariant

embeddings. Augmentations involve corrupting some columns of a sample by drawing from their marginal distributions; (5) STAB (Hajiramezanali et al., 2022): Similar to SimSiam but performs augmentation-free representation learning through stochastic regularization; (6) STUNT (Nam et al., 2023b): Creates self-generated tasks based on clustering to facilitate learning through meta-learning; (7) LFR (Sui et al., 2024): Iteratively learns the target of a pre-text task and the encoder using a random data projector. (8) FeatLLM (Han et al., 2024): Leverages LLMs to discover new rule-based binary features, which are then utilized as representations. (9) MET (Majmundar et al., 2022): Utilizes a Mask-and-Predict approach to uncover the latent structure within the tabular data. Implementation details for all baselines followed the original works, except that the encoder architecture was standardized. Detailed settings can be found in the Appendix C.2. We further compare our model's performance with zero-shot and in-context learning-based inference in Appendix G, though these methods are not designed for unsupervised representation learning. Evaluation results on additional baselines are included in Appendix 4.2.

**Implementation details.** TST-LLM currently employs GPT-3.5 as the LLM backbone for feature discovery but it can be combined with other LLMs. During LLM generation, the temperature was set to 0.5 and the top-p value was set to the API's default of 1. The discovery process generated five features per trial, with the number of trials set at 40. The number of serialized samples included in the prompt was set to a maximum of 20, as allowed by the prompt limit. The number of selected features $M$ was set to 20. Effects of hyper-parameter $M$ are discussed in the section 4.4 and the number of trials is addressed in the Appendix H. The structure of the encoder was consistent with the baselines, configured as a 2-layer MLP with 1024 dimensions, and the projection head consisted of a single linear layer. Effects of the encoder size are discussed in the Appendix I. Training utilized the Adam optimizer with a learning rate of 1e-4, a batch size of 128, and 1000 training iterations. For information on computing resources and computational complexity, refer to the Appendix D.

**Evaluation.** After training, we fixed the learned embeddings, and the evaluation is performed with two downstream task classifiers: (1) Linear model: This can be either logistic regression for classification tasks or linear regression for regression tasks. This method assesses how linearly separable the classes are in the embedding; (2) Non-parametric classifier: This involves fitting a weighted $k$-NN module to the downstream classification task. We run evaluations with two different settings: $k = 3$ and 5. This method evaluates how well the embeddings form coherent local clusters. Evaluation results on additional classifiers are included in Appendix F.2. For performance metrics, AUROC is used for classification tasks (one-versus-all for multi-class settings), and RMSE for regression tasks. Experiments were run with 3 different random seeds, and the average values were reported.

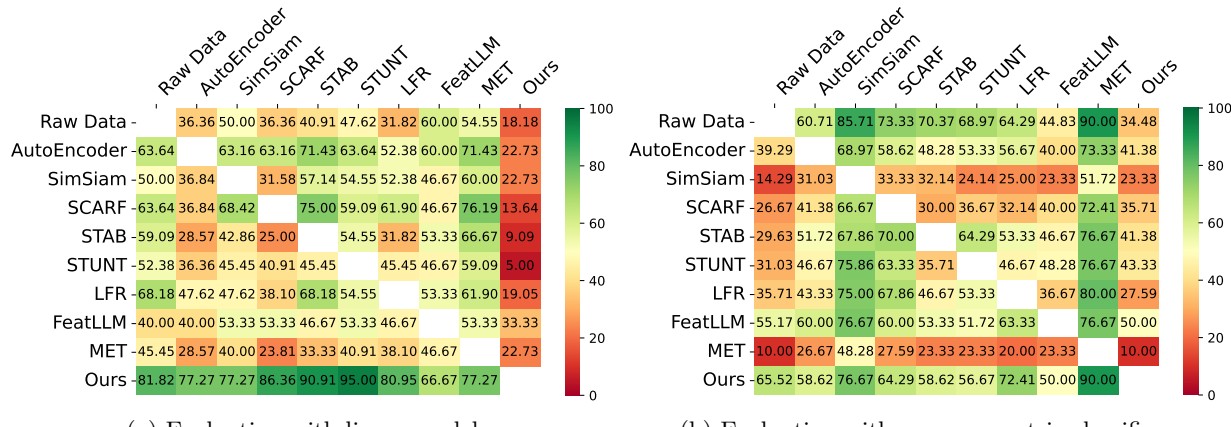

(a) Evaluation with linear model.  (b) Evaluation with non-parametric classifier.

Figure 4: Win matrices comparing self-supervised tabular learning methods against each other with (a) linear model and (b) non-parametric classifier. Self-supervised tabular learning methods are aligned on the x-axis and the y-axis while the numbers represent the winning ratio of the x-axis model against the y-axis model. Full results are reported in the Appendix K.

To facilitate straightforward comparison across datasets, we adopted a win matrix from existing literature (Bahri et al., 2022). The win matrix calculates the ratio over the number of times each method $i$ outperforms another method $j$ across the datasets, excluding ties:

$$W[i,j] = \frac{\sum_{k \in \text{Datasets}} \mathbb{I}[\text{Performance}(i,k) > \text{Performance}(j,k)]}{\sum_{k \in \text{Datasets}} \mathbb{I}[\text{Performance}(i,k) \neq \text{Performance}(j,k)]}, \tag{4}$$

where Performance($i$, $k$) denotes the performance of method $i$ on dataset $k$.

**Results.** Figure 4 compares the performance of self-supervised baselines and TST-LLM against each other using win matrices. For all baselines, the average win ratio is 82% for the linear model and 66% for the non-parametric classifier, demonstrating TST-LLM's superiority; This gives a strong evidence that task-specific pre-text tasks lead to the latent representations that readily form decision boundaries for the target downstream task in the case of the linear model. At the same time, evaluations with a non-parametric classifier indicates that our pre-text tasks effectively extract and utilize information from existing features to enable clustering.

## 4.2 Comparison with Conventional Baselines

In this section, we additionally compare TST-LLM with 8 conventional baselines, including 2 traditional dimension reduction methods and 6 tabular learning methods. (1) PCA (Pearson, 1901): Reduces dimensionality by transforming data into a set of orthogonal components; (2) t-SNE (Van der Maaten & Hinton, 2008): Projects data into a lower-dimensional space preserving local structure; (3) XGBoost (Chen & Guestrin, 2016): Is a gradient-boosting algorithm optimized with parallelization and regularization for high performance. (4) Random Forest (Breiman, 2001): Is an ensemble learning method that builds multiple decision trees and averages their predictions; (5) LightGBM (Ke et al., 2017): Is a gradient-boosting algorithm that handles data using histogram-based learning; (6) Explainable Boosting Machine (EBM, Nori et al. (2019)): Is an interpretable machine learning model that uses generalized additive models with boosting; (7) TabPFN (Hollmann et al., 2023b): Is a zero-shot tabular classification model that predicts outcomes without training by leveraging a pre-trained Transformer; (8) TP-BERTa (Yan et al., 2024): Is a tabular representation learning model with BERT-style pre-training. We set the dimensionality for PCA to 8 and for t-SNE to 2, and we fine-tune TP-BERTa on each dataset prior to comparison. A linear model is used as the downstream task classifier of PCA, t-SNE, and our model.

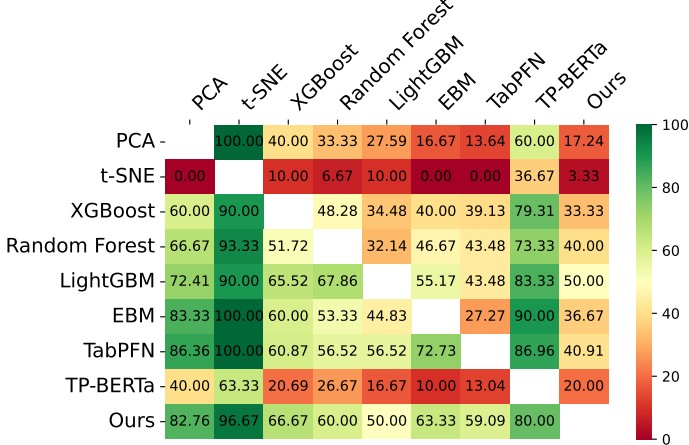

Figure 5: Win matrices comparing conventional baselines with our model. Tabular learning methods are aligned on the x-axis and the y-axis while the numbers represent the winning ratio of the x-axis model against the y-axis model. Full results are reported in the Appendix K.

Figure 5 compares TST-LLM with conventional baselines, across both the datasets used in the main paper and those introduced in Appendix F.1. For all baselines, the average win ratios are 70%, underscoring

TST-LLM's superior performance. Notably, TST-LLM demonstrates results comparable to or exceeding those of supervised tabular learning methods, despite being an unsupervised representation learning framework.

## 4.3 Ablation Study

We conducted an ablation study to evaluate the contribution of each component in our model. We assessed two primary components: discovering features from the downstream task's description and training the encoder with multi-task contrastive learning. We defined the following ablations by removing or modifying each component: (1) Top-1 selection: Only the top-1 feature, which has the least redundancy with the original data among the discovered features, is used; (2) Random-1 selection: Same as Top-1 selection, a single head is used for training, yet the label used for supervised contrastive learning is randomly changed to one of the discovered features in each iteration; (3) Random feature discovery: Instead of using the LLM for feature discovery, we expand features using operations commonly employed in traditional feature engineering work (Zhang et al., 2023), and then randomly select $M$ features. Representation learning is subsequently conducted with these selected features, identical to our original model's approach; (4) Without learning: Instead of performing representation learning, features discovered through feature discovery are directly concatenated with the original data and used as is; (5) Without feature selection: All discovered features are used in representation learning without undergoing the feature selection process.

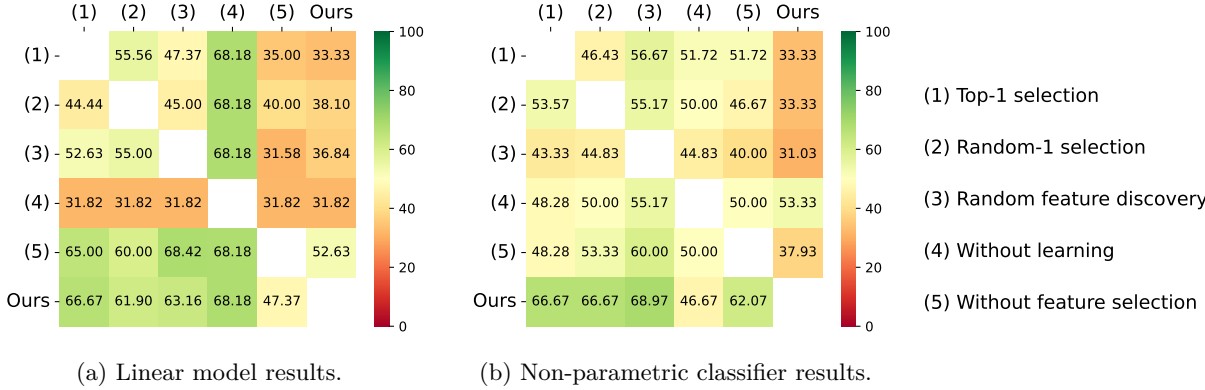

(a) Linear model results.  (b) Non-parametric classifier results.

(1) Top-1 selection

(2) Random-1 selection

(3) Random feature discovery

(4) Without learning

(5) Without feature selection

Figure 6: Win matrices comparing our full model and its ablations against each other with (a) linear model and (b) non-parametric classifier. The numbers represent the winning ratio.

Figure 6 shows the degree of performance degradation in each ablation study. We find that every ablation led to a negative effect on performance, underscoring the contribution of the tested component. Specifically, using multiple features for multi-task learning, rather than relying on a single feature (i.e., Top-1 selection) or alternating features for single-task learning (i.e., Random-1 selection), provided an ensemble effect that enhanced performance. Even without training, merely concatenating features that are relevant to the actual label facilitated the formation of effective local clusters with the non-parametric classifier. By conducting training with a pre-text task, TST-LLM could further obtain embeddings that are linearly separable among the labels. In addition, selecting features does not significantly differ in performance from using all features without feature selection, which suggests that our selection strategy leads to efficient learning (see Section 4.4 for comparison on computational complexity of using all features).

## 4.4 Analysis & Discussion

**How informative are the discovered features for the downstream task?** When training TST-LLM, we utilize the features that have been identified through the LLM. To see how well these pre-text tasks align with actual downstream tasks, we computed the average increase ratio of mutual information between the discovered features and downstream task's labels compared to the original features. The ratio is computed as a percentage for each dataset. According to Figure 7a, for most datasets, the discovered features show a stronger correlation with the labels than the original data. This suggests that our pre-text tasks are

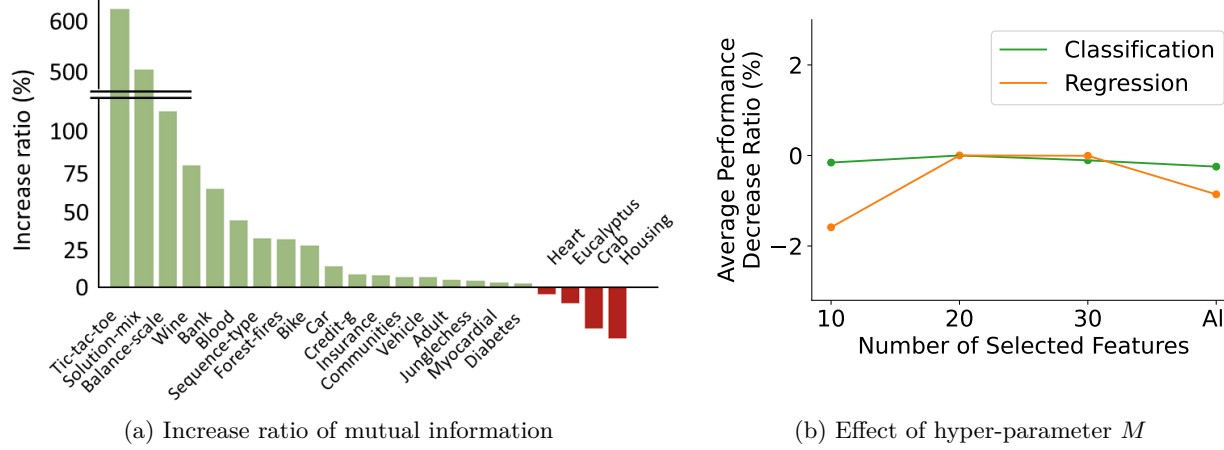

(a) Increase ratio of mutual information

(b) Effect of hyper-parameter $M$

Figure 7: (a) Average increase ratio of mutual information between the discovered features and ground-truth labels compared to the original features. The ratio is reported as a percentage for each dataset; (b) Hyper-parameter analysis on the number of selected features $M$. Average decrease ratios from our model's settings (i.e., $M = 20$) across both classification and regression tasks are reported.

more closely aligned with the actual downstream tasks, than the models trained solely on the original data. We also observed that datasets with a higher increase ratio also demonstrated a greater performance improvement in our model compared to the Raw Data model (Spearman correlation 0.52). When evaluating our model compared to the raw data model over datasets with positive and negative increase ratios, the positive set showed a 25.8% greater improvement in performance (average 8.1% increase in the positive set vs. 5.6% increase in the negative set). Although some datasets showed a decrease in average mutual information, most of them exhibited a high standard deviation in mutual information (see Appendix K.5 for full results), where multi-task learning using a variety of features could be helpful.

**How diverse are the features used for our pre-text task?** We applied two strategies to ensure the diversity of discovered features coming from the LLM and pre-text tasks for representation learning. One strategy involved adding a diversity enforcement component within the LLM's prompt to avoid selecting the previously selected features, and the other aimed to minimize redundancy among selected features. To verify the effectiveness of these methodologies on feature diversity, we conducted additional experiments. We defined three ablation scenarios: (1) No diversity enforcement & No selection strategy: using features without applying the two strategies; (2) No selection strategy: using the diversity enforcement component but not conducting feature selection; (3) With entropy-based filtering: applying only entropy-based filtering as the selection strategy.

Table 1: Ablation study results on feature diversity. Feature diversity is evaluated using the average Cramer's V value across features, with the standard deviation noted. Performance change is computed as an averaged change ratio in percentage across all datasets compared to the proposed full model.

| Ablation for feature diversity | Cramer's V | Performance change (%) | Time cost ratio |
|---|---|---|---|
| No diversity enforcement & No selection strategy | 0.24±0.11 | -0.17±0.24 | 5.62 |
| No selection strategy | 0.13±0.06 | -0.50±0.38 | 4.84 |
| With entropy-based filtering | 0.09±0.05 | -0.05±0.15 | 2.78 |
| Full model | 0.07±0.03 | 0.00±0.00 | 1.00 |

We compared each ablation using three evaluation metrics. The first metric is Cramer's V value (Cramér, 1999) between features, where a higher value indicates a greater number of highly correlated features, implying lower diversity. The second metric is the percentage change in performance across all datasets compared to

the proposed full model. The final metric is the time cost ratio for running the model. According to the results in Table 1, models with lower diversity are inefficient both in terms of performance and time cost.

**Can TST-LLM be applied to other LLM backbones?** To verify whether our method performs well with publicly available LLM backbones beyond GPT-3.5, we applied TST-LLM to the Llama3-7b and Llama3-70b models. Table 2 reports the win ratio of TST-LLM compared to other baselines when applied to different LLM backbones, following the evaluation method in Figure 3-a. The results show that TST-LLM consistently outperforms existing baselines, even with different LLM backbones. Additionally, we observed a positive correlation between the size and reasoning capability of the LLM and the overall model performance. Table 3 reports the average increase ratio of MI between the discovered features and the downstream task's labels compared to the original features, using each LLM backbone. The results indicate that larger LLMs with better reasoning abilities achieve higher MI increase ratios, further supporting the effectiveness of our approach.

Table 2: Comparing TST-LLM with different LLM backbones against self-supervised tabular learning baselines. Each value in the table represents the winning ratio of the row method compared to the column method.

|  | Raw Data | AutoEncoder | SimSiam | SCARF | STAB | STUNT | LFR | FeatLLM | MET | Average |
|---|---|---|---|---|---|---|---|---|---|---|
| Raw Data | 0.00 | 36.36 | 50.00 | 36.36 | 40.91 | 47.62 | 31.82 | 60.00 | 54.55 | 39.74 |
| AutoEncoder | 63.64 | 0.00 | 63.16 | 63.16 | 71.43 | 63.64 | 52.38 | 60.00 | 71.43 | 56.54 |
| SimSiam | 50.00 | 36.84 | 0.00 | 31.58 | 57.14 | 54.55 | 52.38 | 46.67 | 60.00 | 43.24 |
| SCARF | 63.64 | 36.84 | 68.42 | 0.00 | 75.00 | 59.09 | 61.90 | 46.67 | 76.19 | 54.19 |
| STAB | 59.09 | 28.57 | 42.86 | 25.00 | 0.00 | 54.55 | 31.82 | 53.33 | 66.67 | 40.21 |
| STUNT | 52.38 | 36.36 | 45.45 | 40.91 | 45.45 | 0.00 | 45.45 | 46.67 | 59.09 | 41.31 |
| LFR | 68.18 | 47.62 | 47.62 | 38.10 | 68.18 | 54.55 | 0.00 | 53.33 | 61.90 | 48.83 |
| FeatLLM | 40.00 | 40.00 | 53.33 | 53.33 | 46.67 | 53.33 | 46.67 | 0.00 | 53.33 | 42.96 |
| MET | 45.45 | 28.57 | 40.00 | 23.81 | 33.33 | 40.91 | 38.10 | 46.67 | 0.00 | 32.98 |
| Ours (GPT3.5) | 81.82 | 77.27 | 77.27 | 86.36 | 90.91 | 95.00 | 80.95 | 66.67 | 77.27 | **81.50** |
| Ours (Llama3-70B) | 81.82 | 81.82 | 77.27 | 86.36 | 90.91 | 85.71 | 80.00 | 66.67 | 77.27 | **80.87** |
| Ours (Llama3-8B) | 72.73 | 70.00 | 75.00 | 90.00 | 86.36 | 81.82 | 80.00 | 60.00 | 76.19 | **76.90** |

Table 3: Average increase ratio of mutual information across different LLM backbones between the discovered features and ground-truth labels compared to the original features.

| LLM backbone | GPT-3.5 | Llama3-70B | Llama3-8B |
|---|---|---|---|
| Average increase ratio of MI (avg±ste) | 69.99±36.03 | 58.07±20.22 | 47.63±24.00 |

**Does hyper-parameter $M$ affect the performance?** TST-LLM has a hyper-parameter, $M$, which represents the number of features discovered for the pre-text task. To investigate the impact of $M$ on performance, we conducted experiments using $M = 10, 20, 30$, and all features (i.e., $M = $ all) for the pretext task. The results, presented in Figure 7b, include the average decrease ratio in performance from our model's settings across all classification and regression datasets. The performance of TST-LLM is insensitive to $M$ when $M$ is set bigger than 10, allowing for flexibility in choosing the number of features to optimize computational efficiency. Based on our findings, we selected $M = 20$, which delivered the best performance without imposing a computational burden.

## 5 Conclusion

We introduced TST-LLM, a representation learning method that creates pre-text tasks that are tailored to downstream task objectives using an LLM. TST-LLM leverages the prior knowledge and reasoning abilities of the LLM to determine how to combine original data features into informative features based on natural language descriptions of downstream tasks and feature descriptions. The combined features, after undergoing

a feature selection process to minimize redundancy, serve as ground-truth labels for the pre-text tasks in representation learning. Extensive analysis confirms that our methodology can identify diverse and task-aligned features, and as a result consistently achieves outstanding performance across various downstream tasks.

As a potential future direction, we propose extending beyond self-supervised learning to semi-supervised learning by incorporating labeled samples. In situations where meta-information alone is insufficient to define relationships between target labels or where relying on prior knowledge is challenging for specific tasks, labeled samples can serve as a valuable complement. We believe that a model effectively leveraging both the knowledge from LLMs and data can achieve broader applicability.

**Limitations and broader impact.** Our method requires task meta-information for discovery, but this comes at a minimal annotation cost. In addition, our method relies on LLM for feature discovery, which may not yield optimal results for tasks that the LLM is unfamiliar with. To mitigate this, one could consider optimizing alongside traditional self-supervised representation learning objectives in the tabular domain, such as reconstruction (Yoon et al., 2020) or contrastive learning (Bahri et al., 2022). In terms of the impact, TST-LLM facilitates easy learning through task-aligned pre-text tasks with the desired downstream task objective, when these goals can be articulated through text. This adaptability renders it suitable for a variety of real-world scenarios, such as in the healthcare and financial sectors. We believe this work provides a new perspective on the integration of LLMs into the tabular learning domain.

## Acknowledgement

Han, Lee, and Cha were partially supported by the National Research Foundation of Korea grant (RS-2022-00165347).

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

# Appendix

## A   Full Prompt Examples

### A.1   Task Description for Each Dataset

This section presents the downstream task descriptions corresponding to the dataset used for evaluation. TST-LLM uses these text descriptions to perform task-relevant feature discovery. Each description is defined by referencing the dataset's original source or previous works (Hegselmann et al., 2023; Han et al., 2024). For classification tasks, answer class candidates were provided.

Table 4: Downstream task's description of each dataset used for feature discovery.

| Data | Downstream task's description |
|---|---|
| Adult | Does this person earn more than 50000 dollars per year? Yes or no? |
| Balance-scale | Which direction does the balance scale tip to? Right, left, or balanced? |
| Bank | Does this client subscribe to a term deposit? Yes or no? |
| Blood | Did the person donate blood? Yes or no? |
| Car | How would you rate the decision to buy this car? Unacceptable, acceptable, good or very good? |
| Communities | How high will the rate of violent crimes per 100K population be in this area. Low, medium, or high? |
| Credit-g | Does this person receive a credit? Yes or no? |
| Diabetes | Does this patient have diabetes? Yes or no? |
| Eucalyptus | How good is this Eucalyptus species for soil conservation in the specified location? None, low, average, good, or best? |
| Forest-fires | Estimate the burned area of forest fires from given information. |
| Heart | Does the coronary angiography of this patient show a heart disease? Yes or no? |
| Junglechess | Which player wins this two pieces endgame of Jungle Chess? Black, white or draw? |
| Myocardial | Does the myocardial infarction complications data of this patient show chronic heart failure? Yes or no? |
| Tic-tac-toe | Will the first player (player x) win the game? Positive or negative? |
| Vehicle | What kind of vehicle is the given silhouette information about? Bus, opel, saab, or van? |
| Bike | Estimate the count of total rental bikes from given information. |
| Crab | Estimate the age of the crab from given information. |
| Housing | Estimate the house price from given information. |
| Insurance | Estimate the individual medical cost of this patient billed by health insurance. |
| Wine | Estimate the wine quality on a scale from 0 to 10 from given information. |
| Sequence-type | What is the type of following sequence? Arithmetic, geometric, fibonacci, or collatz? |
| Solution-mix | Given the volumes and concentrations of four solutions, calculate the percent concentration of the mixed solution after mixing them. |

## A.2 Full Prompt Example for Feature Discovery

The following is an example of a prompt used for feature discovery on the Adult dataset. For the initial query in the LLM, a prompt without a diversity enforcement component, as shown in Figure 8, was used as there is no information from previous iterations. For subsequent iterations, a prompt with a diversity enforcement component in Figure 9 was used.

---

You are a data engineer. Given the task description and the list of features and data examples, you are making a new column for the data which is informative to solve the task.

Task: Does this person earn more than 50000 dollars per year? Yes or no?
Features:
- age: the age of an individual (numerical variable within range [17, 90])
...
- native-country: country of origin for an individual (categorical variable with categories [United-States, Poland, ..., Holand-Netherlands])

Examples:
age is 49. workclass is Private. fnlwgt is 123807. education is HS-grad. educational-num is 9. marital-status is Separated. occupation is Adm-clerical. relationship is Unmarried. race is Black. gender is Female. capital-gain is 0. capital-loss is 0. hours-per-week is 40. native-country is United-States.
...
age is 52. workclass is Private. fnlwgt is 208302. education is HS-grad. educational-num is 9. marital-status is Married-civ-spouse. occupation is Sales. relationship is Husband. race is White. gender is Male. capital-gain is 0. capital-loss is 0. hours-per-week is 36. native-country is United-States.

Given a type of operations below, generate 5 new columns which are the most informative to solve the task using operations. Refer to the examples when generating features. Only use features listed in the feature description. Note that multiple operations can be nested to generate a new column.

The possible type of operations is as follows:
- Transformations: Numerical features only. Transform the feature value with one of the following operators: absolute, logarithm, square root, sigmoid, or frequency (i.e., frequency of feature in the data).
- Numerical Operations: Numerical features only. Conduct arithmetic operation from multiple columns.
- Mixed-type Operations: Combine categorical feature and numerical feature to generate a new one.
- Categorical Operations: Combine two categorical features to generate a new feature. For example, you can infer a condition to make a binary feature, indicating whether it follows the condition.

Format of response for 5 new columns:
—
Thought 1: [Any reasons based on examples above why the following new feature would be helpful for the task]
New feature 1: [Type of operation] | New_column_name | One line detailed pseudo code for generating columns
...
Thought 5: ...
New feature 5: ...
—

Answer:
—
Thought 1:

---

Figure 8: Full prompt example for feature discovery in the Adult dataset (initial query without diversity enforcement).

You are a data engineer. Given the task description and the list of features and data examples, you are making a new column for the data which is informative to solve the task.

Task: Does this person earn more than 50000 dollars per year? Yes or no?
Features:
- age: the age of an individual (numerical variable within range [17, 90])
...

Examples:
age is 49. workclass is Private. fnlwgt is 123807. education is HS-grad. educational-num is 9. marital-status is Separated. occupation is Adm-clerical. relationship is Unmarried. race is Black. gender is Female. capital-gain is 0. capital-loss is 0. hours-per-week is 40. native-country is United-States.
...

Given a type of operations below, generate 5 new columns which are the most informative to solve the task using operations. Refer to the examples when generating features. Only use features listed in the feature description. Note that multiple operations can be nested to generate a new column.

The possible type of operations is as follows:
- Transformations: Numerical features only. Transform the feature value with one of the following operators: absolute, logarithm, square root, sigmoid, or frequency (i.e., frequency of feature in the data).
- Numerical Operations: Numerical features only. Conduct arithmetic operation from multiple columns.
- Mixed-type Operations: Combine categorical feature and numerical feature to generate a new one.
- Categorical Operations: Combine two categorical features to generate a new feature. For example, you can infer a condition to make a binary feature, indicating whether it follows the condition.

You also have some new example features generated with these modules.

Example Features:
Index | Feature_name | Feature_desc
1 | workclass_gender | Combine workclass and gender to see if certain workclasses have gender-specific income differences
2 | education_marital | Combine education and marital-status to see if higher education levels are correlated with higher income levels
3 | hours_per_week_squared | Square the hours-per-week column to see if there is a non-linear relationship with income
4 | age_log | Take the logarithm of age to normalize the distribution
5 | capital_net | Subtract capital-loss from capital-gain

You must write new feature that is different from all above examples features with respect to both names and descriptions.

Format of response for 5 new columns:
—

Thought 1: [Any reasons based on examples above why the following new feature would be helpful for the task]
New feature 1: [Type of operation] | New_column_name | One line detailed pseudo code for generating columns
...
Thought 5: ...
New feature 5: ...
—

Answer:
—

Thought 1:

Figure 9: Full prompt example for feature discovery in the Adult dataset (subsequent query with diversity enforcement). Text colored blue represents the diversity enforcement component.

### A.3 Example Generation Result from Feature Discovery Prompt

Figure 10 is an example of the outcome when a prompt for feature discovery is input into the LLM. A rationale and a simple description for the implementation of each new feature are generated.

---

Thought 1: Based on the examples, we can see that capital-gain and capital-loss could be important factors in determining if a person earns more than 50000 dollars per year. People with higher capital gains are more likely to earn more.

New feature 1: Categorical Operations | HighEducation | If education is higher than HS-grad, set HighEducation to 1, else 0.

Thought 2: The occupation of an individual can also be indicative of their income level.

New feature 2: Categorical Operations | HighIncomeOccupation | If the occupation is Exec-managerial or Prof-specialty, set HighIncomeOccupation to 1, else 0.

Thought 3: The capital gain and capital loss could be combined to create a net capital feature.

New feature 3: Numerical Operations | NetCapital | NetCapital = capital-gain - capital-loss

Thought 4: The combination of age and hours-per-week could provide insights into the individual's work ethic and experience.

New feature 4: Numerical Operations | WorkEthic | WorkEthic = age * hours-per-week

Thought 5: The combination of marital status and relationship could indicate the individual's family situation, which might affect their income.

New feature 5: Mixed-type Operations | FamilyStatus | If marital-status is Married-civ-spouse and relationship is Husband or Wife, set FamilyStatus to 1, else 0.

---

Figure 10: Example generation result by LLM from feature discovery prompt in the Adult dataset.

## A.4 Full Prompt Example for Parsing Features

Figure 11 shows the prompt used to create converter code for generating actual features based on the new feature descriptions provided by the LLM during the feature discovery process. This prompt is input into the LLM to extract Python-based conversion code for use.

---

Provide me a python code for function, given description below.

Function name: column_appender
Input: Dataframe df_input
Input Features:
- age: the age of an individual (numerical variable within range [17, 90])
...

Output: Dataframe df_output.

Function description: Create a new dataframe df_output. Each column in df_output contains new columns which are made based on the column description below. Be sure that the function code well matches with its feature type (i.e., numerical, categorical), and follows the below guidelines.

Code guidelines (Format - Type | New column name | One line pseudo code):
- Generate each column with one line code.
- If there is no sufficient information to generate code, ignore the corresponding column.
- Make sure that the resulting type column is either string (for categorical) or numeric (for continuous).

Column description:
New feature 1: Categorical Operations | HighEducation | If education is higher than HS-grad, set HighEducation to 1, else 0.

New feature 2: Categorical Operations | HighIncomeOccupation | If the occupation is Exec-managerial or Prof-specialty, set HighIncomeOccupation to 1, else 0.

New feature 3: Numerical Operations | NetCapital | NetCapital = capital-gain - capital-loss

New feature 4: Numerical Operations | WorkEthic | WorkEthic = age * hours-per-week

New feature 5: Mixed-type Operations | FamilyStatus | If marital-status is Married-civ-spouse and relationship is Husband or Wife, set FamilyStatus to 1, else 0.

---

Figure 11: Full prompt example for parsing features in the Adult datsaet.

### A.5 Example Generation Result for Parsing Features Prompt

Figure 12 is an example of the LLM's output for the parsing features prompt. It provides a Python code function for creating the discovered feature.

```
<start>
def column_appender(df_input):

    df_output = df_input.copy()
    df_output['HighEducation'] = df_output['education'].apply(lambda x: 1 if x != 'HS-grad' else 0)
    df_output['HighIncomeOccupation'] = df_output['occupation'].apply(lambda x: 1 if x in
    ['Exec-managerial', 'Prof-specialty'] else 0)
    df_output['NetCapital'] = df_output['capital-gain'] - df_output['capital-loss']
    df_output['WorkEthic'] = df_output['age'] * df_output['hours-per-week']
    df_output['FamilyStatus'] = df_output.apply(lambda x: 1 if (x['marital-status'] == 'Married-
    civ-spouse' and x['relationship'] in ['Husband', 'Wife']) else 0, axis=1)
    return df_output

<end>
```

Figure 12: Example generation results for parsing features prompt in the Adult dataset.

# B  Dataset Details

In our work, we use a total of 22 datasets. Each dataset is assigned to either a binary, multi-class classification, or regression task. The datasets were selected considering the size, variety, and types of features. Basic information of each dataset are shown in Table 5 below. Task objectives of entire datasets are listed in Appendix A.

Table 5: Basic information of datasets used for evaluation.

| Data | # of samples | # of features (Categorical/Numerical) | Task |
|------|------|------|------|
| Adult | 48842 | 14 (7/7) | Binary classification |
| Balance-scale | 625 | 4 (0/4) | Multi-class classification |
| Bank | 45211 | 16 (8/8) | Binary classification |
| Blood | 748 | 4 (0/4) | Binary classification |
| Car | 1728 | 6 (5/1) | Multi-class classification |
| Communities | 1994 | 103 (1/102) | Multi-class classification |
| Credit-g | 1000 | 20 (12/8) | Binary classification |
| Diabetes | 768 | 8 (0/8) | Binary classification |
| Eucalyptus | 736 | 19 (5/14) | Multi-class classification |
| Forest-fires | 517 | 12 (2/10) | Regression |
| Heart | 918 | 11 (4/7) | Binary classification |
| Junglechess | 44819 | 6 (0/6) | Multi-class classification |
| Myocardial | 1700 | 111 (94/17) | Binary classification |
| Tic-tac-toe | 958 | 9 (9/0) | Binary-classification |
| Vehicle | 846 | 18 (0/18) | Multi-class classification |
| Bike | 17379 | 12 (3/9) | Regression |
| Crab | 3893 | 8 (1/7) | Regression |
| Housing | 20640 | 9 (1/8) | Regression |
| Insurance | 1338 | 6 (3/3) | Regression |
| Wine | 6497 | 12 (1/11) | Regression |
| Sequence-type | 250 | 5 (0/5) | Multi-class classification |
| Solution-mix | 300 | 8 (0/8) | Regression |

## C Implementation Details

### C.1 TST-LLM details

This section provides additional implementation details of our model. In the feature discovery process of TST-LLM, we use the GPT-3.5 model as the LLM backbone. Meta-information such as feature names and descriptions were included in the prompt. For categorical features, a list of categories for each feature was added, and for numerical features, the min-max value statistics were included. During LLM generation, the temperature was set to 0.5 and the top-p value was set to the API's default of 1. The discovery process generated five features per trial, with the number of trials set at 40. The number of serialized samples included in the prompt was set to a maximum of 20, as allowed by the prompt limit. When the number of features in a dataset exceeded 100 (e.g., communities, myocardial), and the prompt limit was reached, we resolved this by selecting a random 10 columns per query. Over 40 trials, we ensured that all features were used at least once in the feature discovery process.

After the LLM completed feature discovery, a feature set satisfying the minimum redundancy between the original data was selected for representation learning. The number of selected features $M$ was set to 20. The encoder structure for representation learning was consistent with the baselines, configured as a 2-layer MLP with 1024 dimensions. The projection head consisted of a single linear layer, projecting 1024 to 128 dimensions. Training utilized the Adam optimizer with a learning rate of 1e-4, a batch size of 128, and 1000 training iterations.

### C.2 Baseline details

This section describes the implementation details of the baselines. While the implementation of the baselines followed the original works of the respective papers, the encoder used to extract representations was configured uniformly for a fair comparison (i.e., a 2-layer MLP with 1024 hidden dimensions). Different decoder and projector networks were used according to each methodology.

For Autoencoder baseline, the decoder was the same 2-layer MLP with 1024 hidden dimensions as the encoder. For Siamese network-based methodologies (e.g., SimSiam, SCARF, STAB), a 2-layer MLP with 256 hidden dimensions was used as the projector, and for SimSiam, the predictor consisted of a single linear layer. STUNT, which uses prototype-based learning, does not have a separate decoder. For LFR, a single linear layer predictor and a 2-layer ReLU network with 256 hidden dimensions were used as the random data projector.

For all baselines, we referred to the following links for the implementation[3456].

---

[3]https://github.com/layer6ai-labs/lfr
[4]https://github.com/jaehyun513/STUNT
[5]https://github.com/Sungwon-Han/FeatLLM
[6]https://github.com/google-research/met

# D Computational Complexity

In this section, we compare the computational time required for model training. The comparison was conducted on the Adult dataset using a single A100 GPU. For our model, the computation time includes the entire process of feature discovery and selection from the LLM, as well as training. Table 6 reports the total time spent for each method. We found that our model has a computational time complexity comparable to other baselines.

Table 6: Computational time complexity analysis of self-supervised representation learning methods. The total time spent (in seconds) and the ratio compared to our model are reported for each method.

| Model | Time spent (second) | Time spent (ratio) |
|---|---|---|
| Autoencoder | 520.4 | 1.08 |
| SimSiam | 350.7 | 0.73 |
| SCARF | 479.6 | 1.00 |
| STAB | 208.7 | 0.43 |
| STUNT | 608.8 | 1.27 |
| LFR | 470.3 | 0.98 |
| FeatLLM | 682.2 | 1.42 |
| TST-LLM | 481.2 | 1.00 |

# E  Learning with Other Objectives

In our framework, we utilize supervised contrastive learning to integrate information from LLM-discovered features into embeddings, although it is not the only available approach. Therefore, in this section, we compare the performance of our framework using different loss objectives with a linear model. Figure 13 compares the performance of self-supervised baselines and TST-LLM against each other using win matrices, while our framework uses different training objectives including supervised contrastive learning (Figure 13a), CLIP (Radford et al., 2021) (Figure 13b), reconstruction (Figure 13c), and cross-entropy (Figure 13d). Our framework consistently outperforms other self-supervised baselines, irrespective of the training objectives used.

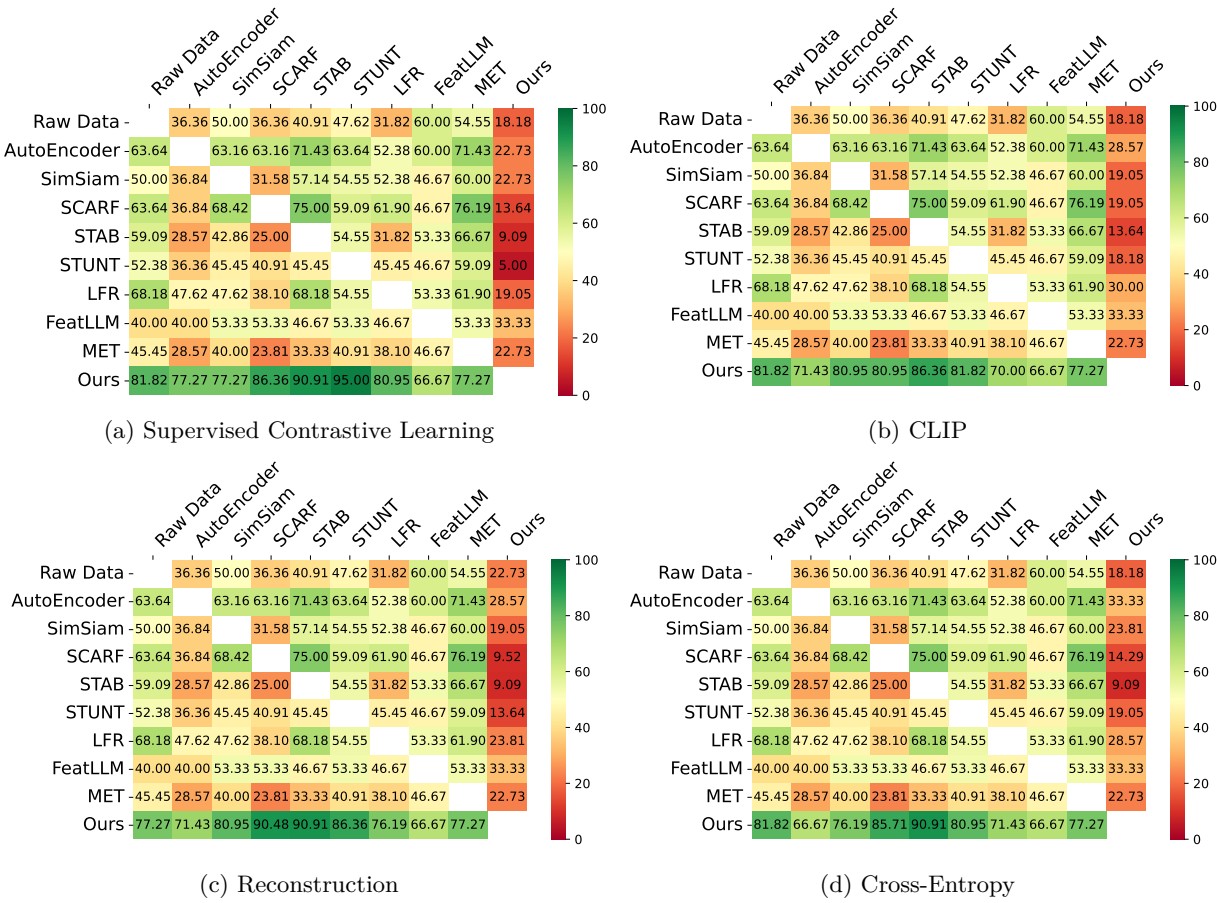

Figure 13: Win matrices comparing self-supervised tabular learning methods against each other, while our framework uses different training objectives including (a) Supervised Contrastive Learning, (b) CLIP, (c) Reconstruction, and (d) Cross-Entropy. Self-supervised tabular learning methods are aligned on the x-axis and the y-axis while the numbers represent the winning ratio of the x-axis model against the y-axis model. Full results are in the Appendix K.7.

# F    Experiments with Additional Datasets & Baselines

## F.1    Comparison on OpenML-CC18 datasets

In this section, we compare TST-LLM with other baseline models using datasets from OpenML-CC18. In addition to the 11 datasets analyzed in the main paper, we have incorporated 8 additional datasets from OpenML-CC18. Additional datasets include metadata about columns and tasks, which are essential for our framework. Detailed information about the additional datasets is provided in Table 7.

Figure 14 displays the comparison results between self-supervised baselines and TST-LLM using win matrices on the OpenML-CC18 datasets. For all baselines, the average win ratios are 83% for the linear model and 64% for the non-parametric classifier, demonstrating TST-LLM's superior performance with the additional datasets.

Table 7: Basic information of additional OpenML-CC18 datasets.

| Data | # of samples | # of features (Categorical/Numerical) | Task |
|---|---|---|---|
| Authorship | 672 | 69 (0/69) | Multi-class classification |
| Breast-w | 559 | 9 (0/9) | Binary classification |
| Cmc | 1178 | 9 (3/6) | Multi-class classification |
| Cylinder-bands | 432 | 39 (19/20) | Binary classification |
| Dmft | 637 | 4 (2/2) | Multi-class classification |
| Ilpd | 466 | 10 (1/9) | Binary classification |
| Optdigits | 4496 | 64 (3/61) | Multi-class classification |
| Pc1 | 887 | 21 (0/21) | Binary classification |

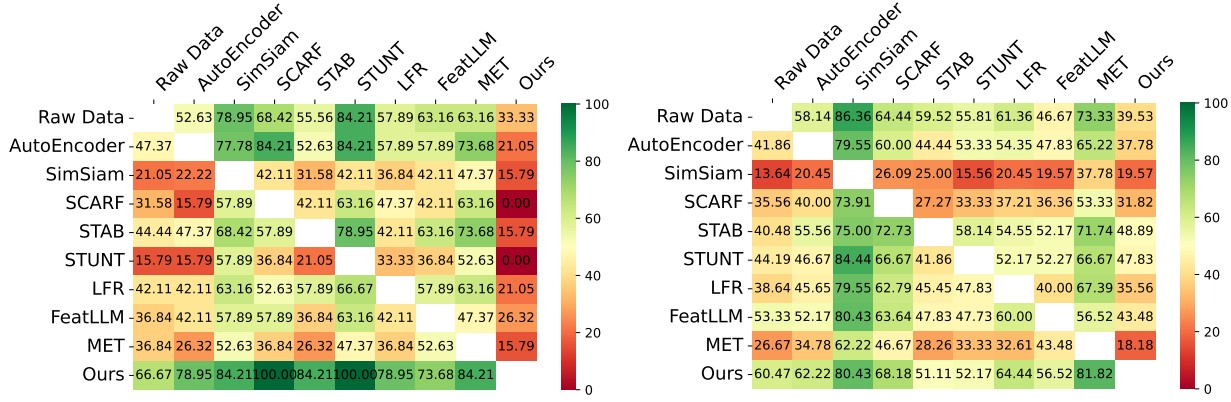

(a) Evaluation with linear model.      (b) Evaluation with non-parametric classifier.

Figure 14: Win matrices comparing self-supervised tabular learning methods against each other with (a) linear model and (b) non-parametric classifier, using OpenML-CC18 datasets. Self-supervised tabular learning methods are aligned on the x-axis and the y-axis while the numbers represent the winning ratio of the x-axis model against the y-axis model. Full results are reported in the Appendix K.

### F.2 Comparison with additional classifiers

In this section, we explore whether TST-LLM offers benefits when applied to other classifiers. We evaluate two tree-based classifiers, XGBoost (Chen & Guestrin, 2016) and Explainable Boosting Machine (EBM, Nori et al. (2019)), alongside two deep learning-based classifiers, Multi-Layer Perceptrons (MLP) and TabNet (Arik & Pfister, 2021).

Table 8 displays the AUC scores for these classifiers using both raw data and the embeddings generated by TST-LLM across 23 classification datasets, as well as the percentage improvement attributed to our model. We can observe that TST-LLM enhances performance in all scenarios except with XGBoost. We hypothesize that this is due to our model being deep learning-based, which likely produces embeddings that facilitate smoother decision boundaries, whereas a tree-based model like XGBoost may perform better with discrete decision boundaries. The performance improvement on other classifiers demonstrates the broad applicability of TST-LLM.

Table 8: Average AUC scores with various classifiers across 23 classification datasets, using the raw data and the embeddings from our model. Full results are reported in the Appendix K.

|  | Linear Model | XGboost | EBM | MLP | TabNet |
|---|---|---|---|---|---|
| Raw Data | 86.25 | 86.95 | 87.35 | 87.01 | 84.08 |
| Ours | 87.74 | 86.51 | 87.81 | 87.02 | 84.16 |
| Improvement Ratio | +1.73% | -0.51% | +0.53% | +0.01% | +0.10% |

# G  Comparison with Zero-Shot & In-Context Learning

TST-LLM performs unsupervised representation learning by targeting features discovered through the LLM. However, leveraging the high generative capabilities of LLMs, one could also consider applying zero-shot inference or in-context learning with few-shot samples for tabular tasks. Table 9 compares the classification performance of our model with zero-shot and in-context learning approaches using the same GPT-3.5 backbone on the Adult dataset. The results demonstrate that our model significantly outperforms both zero-shot and in-context learning methods. Furthermore, we observed that even when using a more advanced model, zero-shot and in-context learning approaches still underperform in prediction tasks compared to our method, underscoring the superiority of our approach.

Table 9: Comparison between TST-LLM and zero-shot / in-context learning-based inference on the Adult dataset. Classification accuracy over the test set is reported.

| GPT-3.5 zero-shot | GPT-3.5 4-shot | GPT-3.5 8-shot | GPT-3.5 16-shot | GPT-4o 16-shot | TST-LLM |
|---|---|---|---|---|---|
| 65.2 | 75.3 | 76.4 | 80.3 | 82.1 | **91.3** |

## H    Impact of the Number of Trials in Feature Discovery

In this section, we analyze the impact of the number of trials on downstream task performance when performing feature discovery through an LLM. In our current model setting, five new features are discovered per trial, and a total of 40 trials are made to obtain the feature set. Figure 15 below measures the performance change ratio compared to the current model as the number of trials is varied to 5, 10, 20, and 30. The results indicate that with 10 or more trials, stable performance is achieved across multiple tasks.

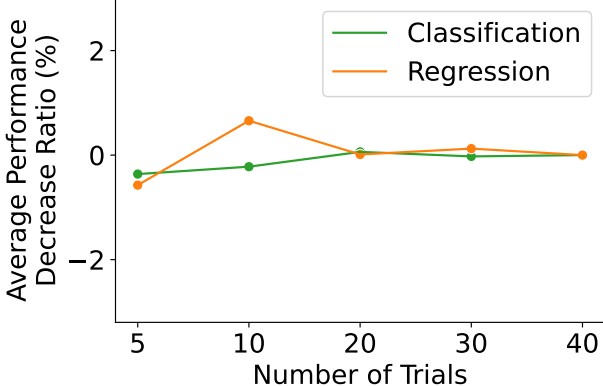

Figure 15: Effect of the number of trials in feature discovery on the performance of downstream tasks.

# I    Impact of the Number of Layers in the Encoder

In this section, we analyze the impact of the number of layers in the encoder on downstream task performance when training the model through supervised contrastive learning. In our default model setting, MLP that consists of two layers with ReLU activation are used. Figure 16 below measures the performance change ratio compared to the default setting as the number of layers in the encoder is varied to 2, 3, 4 and, 5. The results indicate that small number of layers provides better performance, reducing the risk of overfitting.

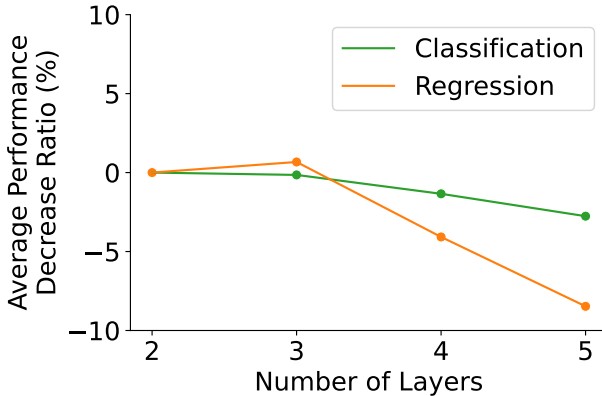

Figure 16: Effect of the number of layers in encoder on the performance of downstream tasks.

## J   Qualitative Analysis

To verify whether the features discovered by the LLM align with the task definition, we selected and examined the top three discovered features for each dataset using our selection strategy (see Table 10). We observed that the discovered features somewhat intuitively align with the downstream task.

Table 10: Top-3 discovered features from our selection strategy for each dataset.

| Data | Top-3 discovered features |
|---|---|
| Adult | age * hours-per-week, educational-num / age, educational-num * age |
| Balance-scale | abs(left-weight - right-weight),
abs(left-weight + left-distance - right-weight - right distance),
(left-weight - left-distance)**2 - (right-weight - right-distance)**2 |
| Bank | duration * campaign, balance * duration, duration / day |
| Blood | (Recency ** 0.5) * Frequency / Time, 1 / (1 + np.exp(Recency - Time)), (Time - Recency) / Frequency |
| Car | maint.map({'low': 1, 'medium': 2, 'high': 3, 'very high': 4}) + doors.map({'5more': 5, '4': 4, '3': 3, '2': 2}),
buying.map({'high':3, 'low':1, 'medium':2, 'very high':4}) + maint.map({'high':3, 'low':1, 'medium':2, 'very high':4})
+ doors.map({'5more':4, '4':3, '2':1, '3':2}) + persons.map({'more':4, '2':1, '4':3})
+ lug_boot.map({'med':2, 'big':3, 'small':1}) + safety.map({'med':2, 'low':1, 'high':3}),
(maint + safety) / 2 |
| Communities | PctEmplManu * HousVacant, MedRentPctHousInc * pctWWage, agePct12t21 * NumInShelters |
| Credit-g | duration / age, age * duration, age / duration |
| Diabetes | Glucose / Age, Pregnancies * DiabetesPedigreeFunction,
DiabetesPedigreeFunction.map(DiabetesPedigreeFunction.value_counts()) |
| Eucalyptus | (Surv + Vig) * Ht, Stem_Fm - Brnch_Fm, Crown_Fm - Brnch_Fm |
| Forest-fires | temp * RH, wind * temp, FFMC + DMC + DC + ISI |
| Heart | Age.corr(MaxHR), RestingBP * MaxHR, abs(RestingBP - MaxHR) |
| Junglechess | (white_piece0_file * white_piece0_rank) / (black_piece0_file * black_piece0_rank),
white_piece0_file * white_piece0_rank,
groupby(['white_piece0_file', 'white_piece0_rank', 'black_piece0_file', 'black_piece0_rank')['white_piece0_file'].transform('count') |
| Myocardial | log(AST_BLOOD), L_BLOOD * ROE, L_BLOOD.value_counts()[L_BLOOD].values |
| Tic-tac-toe | apply(lambda x: (x['top-left-square'] == 'x') + (x['middle-middle-square'] == 'x') + (x['bottom-right-square'] == 'x')),
apply(lambda x: (x['bottom-left-square'] == 'o') + (x['bottom-middle-square'] == 'o') + (x['bottom-right-square'] == 'o')),
apply(lambda x: [x['top-left-square'], x['top-right-square'], x['bottom-left-square'], x['bottom-right-square']].count('o')), |
| Vehicle | COMPACTNESS / CIRCULARITY,
SCALED_RADIUS_OF_GYRATION / RADIUS_RATIO,
PR.AXIS_RECTANGULARITY / CIRCULARITY |
| Bike | abs(temp - hum), abs(temp - atemp), hr * mnth |
| Crab | Shell Weight / (Weight - Shucked Weight - Viscera Weight), Shucked Weight / Viscera Weight, Weight.value_counts() |
| Housing | population / households, total_bedrooms / households, median_income / population |
| Insurance | age * bmi, abs(age - bmi), age / (children + 1) |
| Wine | sulphates - volatile acidity, citric acid / residual sugar, fixed acidity + alcohol |
| Sequence-type | Number2 / Number1 - Number3 / Number2,
['Number1', 'Number2', 'Number3', 'Number4', 'Number5'].sum(axis=1) % 2,
(Number2 / Number1 + Number3 / Number2 + Number4 / Number3 + Number5 / Number4).cumsum() |
| Solution-mix | Solution_1_volume * Solution_1_concentration + Solution_2_volume * Solution_2_concentration + Solution_3_volume
* Solution_3_concentration + Solution_4_volume * Solution_4_concentration,
abs(Solution_1_concentration - Solution_2_concentration) + abs(Solution_2_concentration - Solution_3_concentration)
+ abs(Solution_3_concentration - Solution_4_concentration),
np.log((Solution_1_concentration + Solution_2_concentration + Solution_3_concentration + Solution_4_concentration)
/ (Solution_1_volume + Solution_2_volume + Solution_3_volume + Solution_4_volume)) |

## K   Full Results

In this section, we present full results of our experiments.

### K.1   Evaluation with Linear Model

Table 11: Evaluation results of self-supervised models on linear model, showing (a) AUC across 15 datasets for classification and (b) RMSE across 7 datasets for regression. Best performances are bolded, and our framework's performances, when second-best, are underlined. For FeatLLM, it is not applicable to regression tasks and is therefore denoted as '-'.

(a) Classification (AUC)

| Dataset | Raw Data | AutoEncoder | SimSiam | SCARF | STAB | STUNT | LFR | FeatLLM | MET | Ours |
|---|---|---|---|---|---|---|---|---|---|---|
| Adult | 90.75±0.17 | 91.07±0.20 | 89.01±0.24 | 90.90±0.17 | 90.55±0.24 | 91.06±0.20 | 91.29±0.19 | 88.97±0.49 | 89.98±0.42 | **91.32±0.12** |
| Balance-scale | 97.24±1.11 | 99.58±0.37 | 99.37±0.46 | 99.44±0.39 | 97.66±1.46 | 93.10±2.50 | 99.28±0.39 | **100.00±0.00** | 78.81±1.93 | 99.51±0.33 |
| Bank | 90.48±0.18 | 91.14±0.07 | 87.32±0.16 | 91.73±0.04 | 90.06±0.24 | 91.10±0.38 | 91.65±0.17 | 88.21±0.17 | 90.96±0.16 | **92.08±0.18** |
| Blood | 75.15±3.21 | 74.98±3.52 | 75.18±4.42 | 73.92±4.04 | 74.75±3.24 | 74.39±4.83 | 73.88±3.11 | 66.65±1.07 | **75.87±1.61** | 74.85±2.89 |
| Car | 98.95±0.30 | 99.60±0.23 | 97.95±0.42 | 99.50±0.31 | 99.25±0.43 | 97.96±0.28 | **99.91±0.04** | 99.90±0.04 | 98.43±0.05 | 99.73±0.18 |
| Communities | 84.31±1.23 | 83.01±0.74 | 83.56±0.85 | 83.86±1.51 | 84.39±0.70 | 85.09±1.15 | 81.45±1.12 | 81.77±0.24 | 71.73±5.10 | **85.25±0.67** |
| Credit-g | 77.89±6.44 | 77.60±5.26 | 77.69±6.27 | 77.12±5.46 | 77.26±3.18 | 75.94±4.51 | 75.04±6.32 | 77.40±1.57 | 75.45±1.77 | **78.38±4.85** |
| Diabetes | 83.07±4.74 | 81.64±6.16 | 82.73±5.58 | 81.96±5.97 | 80.38±5.22 | 82.21±3.04 | 81.43±7.38 | 83.43±1.94 | **86.15±2.02** | 82.56±5.12 |
| Eucalyptus | **91.64±1.10** | 90.85±1.33 | 90.50±1.80 | 90.44±1.23 | 89.66±1.97 | 85.61±1.51 | 89.85±0.64 | 90.83±0.02 | 85.46±1.10 | 91.34±0.99 |
| Heart | 93.10±2.12 | 92.79±1.60 | 93.07±2.33 | 93.15±1.58 | 93.15±2.52 | 92.38±2.70 | 92.60±2.16 | 92.50±1.53 | 93.07±2.00 | **93.45±1.60** |
| Junglechess | 80.61±0.33 | 89.89±0.49 | 86.92±0.70 | 88.45±0.70 | 92.10±0.47 | 91.62±0.44 | 92.93±0.42 | 90.60±0.73 | 85.46±0.19 | **93.43±0.31** |
| Myocardial | 61.20±5.13 | 60.90±4.97 | 66.11±4.05 | 60.43±3.35 | 59.29±3.86 | 63.27±4.35 | 62.06±3.38 | 57.01±1.18 | **68.23±2.29** | 63.64±3.08 |
| Sequence-type | 92.11±2.03 | 96.37±0.75 | 96.34±1.17 | 97.36±0.91 | 97.16±1.48 | 92.40±1.15 | 97.41±0.63 | **100.00±0.00** | 93.02±1.22 | 96.44±1.01 |
| Tic-tac-toe | 99.31±0.60 | 99.84±0.08 | 98.28±1.35 | 99.00±0.67 | 95.93±1.87 | 94.07±3.14 | 99.80±0.15 | **100.00±0.00** | 95.80±2.17 | 99.52±0.51 |
| Vehicle | 94.82±0.50 | 96.16±0.83 | 92.37±1.39 | 96.02±0.52 | 95.32±0.49 | 93.55±1.07 | **96.32±0.38** | 89.20±0.25 | 92.11±1.12 | 96.22±0.28 |

(b) Regression (RMSE)

| Dataset | Raw Data | AutoEncoder | SimSiam | SCARF | STAB |
|---|---|---|---|---|---|
| Bike | 142.36±1.58 | 126.90±1.02 | 121.59±1.59 | 111.67±2.08 | 126.42±1.91 |
| Crab | 2.21±0.05 | **2.12±0.03** | **2.12±0.04** | **2.12±0.03** | 2.15±0.02 |
| Forest-fires | **75.07±35.28** | 81.21±29.08 | 82.01±27.45 | 82.87±28.15 | 80.13±30.24 |
| Housing | 69132.79±489.67 | 58155.43±619.72 | 59159.48±62.79 | 56941.63±519.79 | 60071.46±297.95 |
| Insurance | 5930.14±273.29 | 4641.78±220.29 | 4666.29±252.95 | 4657.87±174.01 | 4787.10±170.53 |
| Solution-mix | 0.07±0.00 | 0.03±0.00 | 0.03±0.00 | 0.03±0.00 | 0.03±0.00 |
| Wine | 0.73±0.01 | 0.69±0.00 | 0.69±0.00 | 0.69±0.01 | 0.71±0.00 |

| Dataset | STUNT | LFR | FeatLLM | MET | Ours |
|---|---|---|---|---|---|
| Bike | 115.98±2.18 | 121.02±2.43 | - | **87.83±0.92** | 111.46±1.72 |
| Crab | 2.13±0.04 | 2.16±0.02 | - | 2.15±0.04 | 2.13±0.03 |
| Forest-fires | 77.57±32.56 | 83.84±26.81 | - | 76.36±28.56 | 83.19±22.47 |
| Housing | 56151.34±352.44 | 58064.28±312.73 | - | 60230.45±864.27 | **56069.83±406.99** |
| Insurance | 5099.03±337.72 | 4833.99±293.63 | - | 6913.13±189.72 | **4578.14±149.83** |
| Solution-mix | 0.07±0.00 | **0.02±0.00** | - | 0.08±0.00 | **0.02±0.01** |
| Wine | **0.67±0.00** | 0.69±0.01 | - | 0.69±0.01 | **0.67±0.00** |

## K.2 Evaluation with Non-Parametric Classifier

Table 12: Evaluation results of self-supervised models on Non-parametric classifier with (a) 3 and (b) 5 clusters, showing AUC across 16 datasets for classification. Best performances are bolded, and our framework's performances, when second-best, are underlined.

(a) 3 Clusters

| Dataset | Raw Data | AutoEncoder | SimSiam | SCARF | STAB | STUNT | LFR | FeatLLM | MET | Ours |
|---|---|---|---|---|---|---|---|---|---|---|
| Adult | **82.31±0.18** | 81.53±0.16 | 80.51±0.41 | 81.90±0.09 | 82.21±0.24 | 82.20±0.30 | 82.28±0.29 | 81.63±0.17 | 80.81±0.29 | 81.90±0.46 |
| Balance-scale | 79.47±2.44 | 78.40±1.60 | 82.67±1.85 | 78.67±0.46 | 79.73±3.23 | 79.20±1.60 | 78.13±2.44 | **98.80±1.70** | 65.87±3.03 | 79.73±0.46 |
| Bank | 89.09±0.14 | 89.19±0.03 | 88.41±0.25 | 88.85±0.13 | 89.11±0.04 | 88.96±0.11 | 89.16±0.11 | 87.53±0.08 | 88.34±0.22 | **89.36±0.26** |
| Blood | 72.44±4.44 | 71.33±3.46 | 71.33±4.67 | 69.78±5.00 | 71.33±4.67 | **74.67±3.71** | 72.00±4.16 | 72.67±1.89 | 70.44±2.78 | 74.33±3.06 |
| Car | 87.09±2.09 | 86.42±2.37 | 77.17±1.16 | 79.00±0.73 | 82.01±1.59 | 82.85±1.97 | 79.77±2.08 | 81.36±0.61 | 88.92±6.46 | **89.40±2.73** |
| Communities | 63.07±2.25 | 62.32±2.43 | 58.23±1.61 | 61.99±1.63 | 61.57±0.95 | 62.24±1.04 | 62.91±0.90 | **64.29±0.89** | 55.72±2.85 | 62.16±2.39 |
| Credit-g | 73.00±1.50 | **73.33±0.76** | 69.00±4.44 | 71.83±3.33 | 72.00±3.04 | 71.67±4.65 | 71.50±1.00 | 73.00±0.00 | 70.50±1.32 | 71.83±1.76 |
| Diabetes | 73.16±4.96 | 72.94±6.57 | 69.91±7.30 | 70.78±3.95 | **74.24±1.35** | 74.03±4.06 | 72.51±4.12 | 74.03±2.75 | 71.43±1.72 | 72.73±4.90 |
| Eucalyptus | 59.23±3.19 | 53.60±3.96 | 57.88±2.73 | 52.03±4.05 | 56.08±4.22 | 52.25±2.06 | 58.23±2.06 | **61.82±1.43** | 58.28±5.87 | 60.59±5.12 |
| Heart | 84.60±1.66 | 84.96±1.91 | 84.42±1.13 | 85.33±0.54 | **85.51±1.75** | 85.05±2.57 | 84.60±2.45 | 85.14±2.74 | 85.14±2.74 | 84.70±1.09 |
| Junglechess | 75.08±0.54 | 74.35±0.27 | **77.40±0.14** | 72.34±0.22 | 74.84±0.64 | 73.65±0.47 | 73.87±0.38 | 72.31±0.65 | 72.31±0.65 | 75.35±0.52 |
| Myocardial | 74.40±5.63 | 73.91±2.51 | 73.43±2.93 | **75.12±1.11** | 73.43±1.11 | 71.01±3.32 | 74.64±4.35 | 71.29±5.67 | 71.29±5.67 | 72.22±2.74 |
| Sequence-type | 90.00±2.00 | 91.33±1.15 | 86.00±3.46 | **93.33±1.15** | 90.67±2.31 | 92.00±2.00 | 93.33±2.31 | 84.67±6.11 | 84.67±6.11 | 91.33±2.31 |
| Tic-tac-toe | 91.15±0.52 | 82.29±0.90 | 85.07±2.41 | 72.40±2.71 | 84.90±1.56 | 96.70±1.59 | 77.95±2.46 | 86.11±4.37 | 86.11±4.37 | **93.92±1.59** |
| Vehicle | 69.80±2.96 | **75.10±3.55** | 59.80±2.96 | 68.82±5.23 | 68.80±1.80 | 67.59±3.67 | 74.31±4.34 | 63.51±4.44 | 63.51±4.44 | 74.51±2.65 |

(b) 5 Clusters

| Dataset | Raw Data | AutoEncoder | SimSiam | SCARF | STAB | STUNT | LFR | FeatLLM | MET | Ours |
|---|---|---|---|---|---|---|---|---|---|---|
| Adult | 83.17±0.19 | 82.48±0.12 | 81.47±0.31 | 82.60±0.11 | 83.17±0.11 | 82.93±0.35 | 83.15±0.37 | 81.65±0.22 | 81.53±0.34 | **83.22±0.32** |
| Balance-scale | 82.40±2.88 | 82.40±2.12 | 86.67±0.46 | 83.20±2.40 | 81.60±3.20 | 85.07±2.81 | 81.33±3.33 | **98.00±2.83** | 69.87±5.90 | 81.33±0.92 |
| Bank | 89.40±0.32 | **89.48±0.14** | 88.85±0.22 | 89.13±0.16 | 89.54±0.04 | 89.37±0.26 | 89.47±0.27 | 88.08±0.10 | 89.05±0.17 | 89.25±0.06 |
| Blood | **74.67±4.16** | 74.44±4.07 | 73.56±4.73 | 74.44±2.78 | **74.67±2.91** | 74.22±4.34 | 72.89±3.67 | 73.67±4.24 | 71.78±2.69 | 73.78±4.07 |
| Car | 89.69±0.83 | 84.90±1.01 | 78.13±1.77 | 85.07±1.64 | 88.54±2.53 | 88.54±2.67 | 84.78±1.92 | 83.09±0.20 | 91.33±4.26 | **93.77±2.29** |
| Communities | 65.58±1.16 | 64.55±1.24 | 61.32±0.14 | 63.16±1.57 | 63.41±1.96 | 65.25±1.67 | 64.91±1.15 | **67.29±1.24** | 56.47±2.54 | 65.58±1.43 |
| Credit-g | 72.33±2.57 | **74.83±0.29** | 71.17±4.54 | 73.50±3.61 | 71.83±2.93 | 71.33±1.15 | 71.67±1.53 | 70.00±2.12 | 70.50±3.61 | 73.17±3.88 |
| Diabetes | 73.38±3.25 | 72.51±4.32 | 73.38±5.15 | 73.16±2.46 | 72.94±3.33 | **74.03±3.62** | 73.38±3.90 | 73.70±4.13 | 73.16±3.20 | 71.65±5.52 |
| Eucalyptus | 59.46±4.87 | 54.95±3.96 | 58.33±2.56 | 53.60±3.47 | 54.28±5.46 | 52.70±3.10 | 58.33±3.96 | 60.14±2.87 | 56.01±7.27 | **63.16±3.10** |
| Heart | 85.69±1.91 | 86.23±0.31 | 85.69±1.57 | 86.41±1.44 | 84.96±0.63 | 85.69±0.83 | **86.59±1.66** | 81.25±4.23 | 85.33±3.56 | 84.06±1.37 |
| Junglechess | 75.20±0.42 | 75.28±0.40 | **78.93±0.57** | 74.04±0.54 | 76.09±0.61 | 75.57±0.33 | 75.23±0.52 | 75.97±1.62 | 72.99±0.51 | 75.80±0.43 |
| Myocardial | 75.60±2.74 | 75.85±1.82 | 73.91±0.72 | 76.33±1.11 | **76.57±1.82** | 74.40±2.54 | 76.09±2.90 | 77.54±1.02 | 75.18±1.93 | 76.12±2.33 |
| Sequence-type | 91.33±3.06 | 91.33±1.15 | 86.00±5.29 | 93.33±2.31 | 90.00±3.46 | 90.00±4.00 | 93.33±2.31 | **100.00±0.00** | 86.00±8.72 | 92.67±3.06 |
| Tic-tac-toe | 94.10±0.80 | 84.38±1.80 | 87.33±1.97 | 77.26±2.46 | 90.45±2.87 | 97.74±0.60 | 82.12±1.08 | **100.00±0.00** | 90.28±4.37 | 95.31±1.88 |
| Vehicle | 72.75±1.70 | 75.49±0.90 | 60.39±0.34 | 70.39±3.59 | 72.75±0.34 | 71.57±2.23 | 74.71±3.11 | 67.65±5.82 | 62.13±5.33 | **76.47±3.53** |

### K.3 Ablation Study with Linear Model

Table 13: Evaluation results of ablation studies on linear model, showing (a) AUC across 15 datasets for classification and (b) RMSE across 7 datasets for regression. Best performances are bolded, and our framework's performances, when second-best, are underlined.

(a) Classification (AUC)

| Dataset | Top-1 selection | Random-1 selection | Random feature discovery | Without learning | Without feature selection | Ours |
|---|---|---|---|---|---|---|
| Adult | 91.27±0.11 | 91.27±0.13 | **91.38±0.11** | 91.65±0.54 | 91.36±0.13 | 91.32±0.12 |
| Balance-scale | 99.46±0.37 | 99.53±0.29 | 99.54±0.26 | **99.96±0.07** | 99.64±0.26 | 99.51±0.33 |
| Bank | **92.10±0.12** | 91.96±0.20 | 91.99±0.24 | 89.83±0.65 | 91.91±0.19 | 92.08±0.18 |
| Blood | 74.72±2.85 | 74.89±2.83 | 74.85±2.85 | 73.12±4.94 | **75.26±2.67** | 74.85±2.89 |
| Car | 99.65±0.22 | 99.48±0.40 | 99.68±0.16 | 98.05±1.75 | **99.81±0.08** | 99.73±0.18 |
| Communities | 85.37±1.28 | 84.74±0.44 | 85.36±0.73 | 84.19±1.46 | **85.59±1.28** | 85.25±0.67 |
| Credit-g | 78.23±5.14 | 77.97±5.89 | 77.53±5.48 | 74.48±4.54 | **78.43±4.46** | 78.38±4.85 |
| Diabetes | 82.20±5.36 | 82.23±5.19 | **82.60±4.81** | 84.33±3.89 | 82.19±5.20 | 82.56±5.12 |
| Eucalyptus | 91.16±0.84 | 90.92±1.26 | 91.15±0.89 | 88.94±0.80 | **91.41±1.13** | 91.34±0.99 |
| Heart | **93.56±1.44** | 93.46±1.28 | 93.25±1.47 | 92.50±1.63 | 93.47±1.48 | 93.45±1.60 |
| Junglechess | 93.37±0.28 | 92.07±0.29 | 93.37±0.21 | **93.57±1.54** | 93.43±0.39 | 93.43±0.31 |
| Myocardial | 62.10±2.45 | 63.05±1.51 | 62.46±2.78 | **63.98±2.70** | 62.23±3.70 | 63.64±3.08 |
| Sequence-type | 96.58±0.90 | **96.80±1.02** | 96.45±1.01 | 83.33±28.87 | 96.47±1.10 | 96.44±1.01 |
| Tic-tac-toe | 99.58±0.23 | 99.24±0.42 | 98.95±0.53 | **99.95±0.06** | 99.47±0.33 | 99.52±0.51 |
| Vehicle | 95.95±0.51 | 95.94±0.47 | 96.00±0.56 | 94.69±0.30 | 96.08±0.53 | **96.22±0.28** |

(b) Regression (RMSE)

| Dataset | Top-1 selection | Random-1 selection | Random feature discovery | Without learning | Without feature selection | Ours |
|---|---|---|---|---|---|---|
| Bike | 112.70±1.83 | 111.27±1.83 | **111.09±1.75** | 740.74±657.86 | 111.68±2.18 | 111.46±1.72 |
| Crab | **2.12±0.03** | **2.12±0.01** | 2.13±0.02 | 6.94±6.20 | 2.13±0.02 | 2.13±0.03 |
| Forest-fires | 81.61±22.83 | **81.41±23.53** | 83.55±23.47 | 87.11±32.62 | 86.03±24.27 | 83.19±22.47 |
| Housing | 56162.70±247.14 | 56231.24±334.36 | 55984.82±323.18 | 65521.90±9492.64 | **55864.26±90.89** | 56069.83±406.99 |
| Insurance | 4630.87±139.19 | **4567.01±116.92** | 4587.91±118.33 | 5880.78±1411.86 | 4582.00±179.75 | 4578.14±149.83 |
| Solution-mix | 0.02±0.01 | 0.02±0.00 | 0.02±0.00 | **0.01±0.00** | 0.02±0.00 | 0.02±0.01 |
| Wine | 0.68±0.00 | 0.68±0.00 | 0.68±0.00 | 0.90±0.26 | 0.68±0.01 | **0.67±0.00** |

### K.4 Ablation Study with Non-parametric Classifier

Table 14: Evaluation results of ablation studies on non-parametric classifier with (a) 3 and (b) 5 clusters across 16 datasets for classification. Best performances are bolded, and our framework's performances, when second-best, are underlined.

(a) 3 Clusters

| Dataset | Top-1 selection | Random-1 selection | Random feature discovery | Without learning | Without feature selection | Ours |
|---|---|---|---|---|---|---|
| Adult | 81.87±0.41 | 82.11±0.08 | 82.64±0.36 | **83.07±0.23** | 83.04±0.21 | 81.90±0.46 |
| Balance-scale | 73.07±5.45 | 79.20±1.39 | 80.00±1.60 | **86.13±0.46** | 80.27±0.92 | 79.73±0.46 |
| Bank | 89.08±0.05 | 89.04±0.10 | 88.91±0.36 | 88.76±0.29 | 88.71±0.19 | **89.36±0.26** |
| Blood | 71.33±1.76 | 72.44±5.05 | 73.33±3.06 | **76.00±2.00** | 72.67±3.53 | 74.33±3.06 |
| Car | 82.56±9.62 | 89.02±2.37 | 82.66±6.95 | 81.79±5.69 | 83.82±3.33 | **89.40±2.73** |
| Communities | 63.32±2.27 | 64.24±2.04 | 62.74±0.29 | 63.24±1.24 | **64.75±1.16** | 62.16±2.39 |
| Credit-g | 70.00±2.60 | 71.33±2.57 | **72.33±1.76** | 72.00±5.63 | 71.00±2.65 | 71.83±1.76 |
| Diabetes | 72.94±5.52 | 72.29±4.32 | 71.94±4.92 | **73.38±5.15** | 72.94±6.14 | 72.73±4.90 |
| Eucalyptus | 56.53±3.96 | 59.68±2.73 | 58.11±1.17 | 56.53±2.81 | **62.39±4.50** | 60.59±5.12 |
| Heart | 85.69±1.75 | 84.24±2.72 | 84.05±1.57 | **86.78±0.83** | 84.06±2.57 | 84.70±1.09 |
| Junglechess | 75.46±1.21 | 75.45±0.75 | 74.91±0.63 | **75.76±2.05** | 74.15±0.52 | 75.35±0.52 |
| Myocardial | **75.12±4.25** | 73.19±3.32 | 73.67±0.84 | 72.46±4.35 | 74.15±2.33 | 72.22±2.74 |
| Sequence-type | 90.00±3.46 | 89.33±2.31 | 91.33±2.31 | 73.33±29.14 | **92.00±3.46** | 91.33±2.31 |
| Tic-tac-toe | 90.80±1.20 | 91.49±1.50 | 81.25±5.29 | 91.15±12.18 | **95.49±1.31** | 93.92±1.59 |
| Vehicle | 71.76±5.79 | 70.59±5.79 | 72.75±4.42 | 74.12±0.59 | 70.78±1.48 | **74.51±2.65** |

(b) 5 Clusters

| Dataset | Top-1 selection | Random-1 selection | Random feature discovery | Without learning | Without feature selection | Ours |
|---|---|---|---|---|---|---|
| Adult | 82.64±0.15 | 82.90±0.13 | 83.43±0.35 | **84.03±0.11** | 83.83±0.22 | 83.22±0.32 |
| Balance-scale | 81.33±3.23 | 82.93±2.44 | 82.93±1.22 | **86.13±2.44** | 81.07±2.01 | 81.33±0.92 |
| Bank | **89.51±0.16** | 89.37±0.13 | 89.21±0.20 | 89.41±0.18 | 89.25±0.08 | 89.25±0.06 |
| Blood | 73.78±2.14 | 75.33±3.06 | 73.56±3.79 | **76.00±3.06** | 72.67±2.91 | 73.78±4.07 |
| Car | 88.82±6.79 | 91.04±2.00 | 88.54±3.47 | 83.62±5.43 | 89.69±0.60 | **93.77±2.29** |
| Communities | 65.41±1.15 | 65.33±1.43 | 64.41±1.25 | 64.24±1.13 | 64.83±2.03 | **65.58±1.43** |
| Credit-g | 73.17±1.04 | **73.33±2.36** | 73.00±2.60 | 71.33±5.53 | 71.67±3.62 | 73.17±3.88 |
| Diabetes | 73.59±5.67 | 73.59±3.33 | **74.24±4.92** | 72.29±4.61 | 73.16±4.70 | 71.65±5.52 |
| Eucalyptus | 58.56±7.29 | 59.46±5.41 | 56.76±4.11 | 58.11±3.76 | 62.39±6.39 | **63.16±3.10** |
| Heart | 85.69±1.91 | 85.14±1.66 | **87.14±1.13** | 85.87±0.54 | 84.24±1.44 | 84.06±1.37 |
| Junglechess | 76.51±1.35 | 75.30±0.31 | 75.73±0.60 | **76.94±1.88** | 75.05±0.41 | 75.80±0.43 |
| Myocardial | 75.85±1.67 | 73.91±0.72 | 75.36±1.92 | 75.36±2.90 | 75.60±1.11 | **76.12±2.33** |
| Sequence-type | 91.33±4.16 | 91.33±4.16 | 90.67±4.62 | 73.33±29.48 | 92.00±4.00 | **92.67±3.06** |
| Tic-tac-toe | 94.62±0.60 | 92.53±2.10 | 86.98±5.02 | 91.15±11.30 | 93.92±1.59 | **95.31±1.88** |
| Vehicle | 72.75±2.78 | 73.14±3.02 | 74.51±1.36 | 70.59±4.08 | 73.33±1.80 | **76.47±3.53** |

### K.5 Informativeness of Discovered Features

Table 15: Analysis of the informativeness of features discovered via LLM. The average mutual information (MI) between features and the downstream task's labels is reported for each dataset. The increase ratio in MI when using discovered features compared to original features is also reported, along with standard deviations.

| Data | Average MI in original features | Average MI in discovered features | Increase ratio (%) |
|---|---|---|---|
| Tic-tac-toe | 0.010 | 0.076 | 646.6±1535.1 |
| Solution-mix | 0.064 | 0.386 | 507.3±1348.2 |
| Balance-scale | 0.082 | 0.178 | 117.3±299.0 |
| Wine | 0.055 | 0.100 | 81.2±112.3 |
| Bank | 0.013 | 0.022 | 65.8±172.0 |
| Blood | 0.031 | 0.045 | 44.6±95.2 |
| Sequence-type | 0.345 | 0.459 | 32.9±86.9 |
| Forest-fires | 0.019 | 0.025 | 32.2±131.3 |
| Bike | 0.103 | 0.132 | 27.9±155.0 |
| Car | 0.036 | 0.041 | 14.2±157.5 |
| Credit-g | 0.009 | 0.009 | 8.8±136.5 |
| Insurance | 0.364 | 0.395 | 8.4±127.7 |
| Communities | 0.089 | 0.095 | 7.1±85.2 |
| Vehicle | 0.212 | 0.226 | 6.8±56.5 |
| Adult | 0.031 | 0.032 | 5.3±123.1 |
| Junglechess | 0.049 | 0.051 | 4.8±73.8 |
| Myocardial | 0.007 | 0.007 | 3.1±95.5 |
| Diabetes | 0.043 | 0.045 | 3.0±71.9 |
| Heart | 0.067 | 0.063 | -5.1±81.5 |
| Eucalyptus | 0.177 | 0.158 | -10.9±86.9 |
| Crab | 0.350 | 0.254 | -27.5±43.7 |
| Housing | 0.154 | 0.102 | -34.0±72.2 |

### K.6 Hyperparameter Analysis on $M$

Table 16: Evaluation results with various hyperparameter $M$ on linear model, showing (a) AUC across 15 datasets for classification and (b) RMSE across 7 datasets for regression. Best performances are bolded.

(a) Classification (AUC)

| Dataset | $M = 10$ | $M = 20$ | $M = 30$ | $M = $ All |
|---|---|---|---|---|
| Adult | **91.33±0.16** | 91.32±0.12 | 91.31±0.14 | 91.27±0.16 |
| Balance-scale | 99.50±0.36 | 99.51±0.33 | 99.57±0.31 | **99.62±0.28** |
| Bank | 92.05±0.23 | **92.08±0.18** | 92.01±0.24 | 91.84±0.06 |
| Blood | **74.95±2.95** | 74.85±2.89 | **74.95±2.80** | 74.72±3.04 |
| Car | 99.77±0.13 | 99.73±0.18 | **99.80±0.13** | 99.78±0.14 |
| Communities | 85.30±1.28 | 85.25±0.67 | **85.70±0.88** | 85.25±0.89 |
| Credit-g | **79.07±5.94** | 78.38±4.85 | 78.66±5.36 | 78.08±4.92 |
| Diabetes | 81.86±5.34 | **82.56±5.12** | 81.99±5.31 | 82.02±4.71 |
| Eucalyptus | 91.58±0.81 | 91.34±0.99 | 91.49±0.88 | **91.74±0.45** |
| Heart | **93.69±1.26** | 93.45±1.60 | 93.52±1.58 | 93.42±1.49 |
| Junglechess | 93.29±0.29 | 93.43±0.31 | **93.64±0.36** | 93.45±0.33 |
| Myocardial | 61.72±2.65 | **63.64±3.08** | 61.95±3.44 | 61.77±2.43 |
| Sequence-type | 96.50±0.92 | 96.44±1.01 | **96.69±0.97** | 96.61±0.91 |
| Tic-tac-toe | **99.67±0.36** | 99.52±0.51 | 99.59±0.41 | 99.45±0.56 |
| Vehicle | 96.08±0.46 | **96.22±0.28** | 96.12±0.52 | 96.15±0.52 |

(b) Regression (RMSE)

| Dataset | $M = 10$ | $M = 20$ | $M = 30$ | $M = $ All |
|---|---|---|---|---|
| Bike | 112.57±1.73 | 111.46±1.72 | **110.71±2.62** | 111.09±1.29 |
| Crab | **2.13±0.02** | **2.13±0.03** | **2.13±0.01** | 2.14±0.03 |
| Forest-fires | 83.33±23.21 | 83.19±22.47 | 82.17±22.78 | **81.14±23.21** |
| Housing | 56266.21±248.64 | 56069.83±406.99 | **56047.02±125.19** | 56056.16±240.30 |
| Insurance | 4622.89±173.26 | 4578.14±149.83 | 4615.54±160.49 | **4614.74±196.02** |
| Solution-mix | **0.02±0.00** | **0.02±0.01** | **0.02±0.00** | **0.02±0.00** |
| Wine | 0.68±0.01 | **0.67±0.00** | 0.68±0.00 | 0.68±0.00 |

### K.7 Learning with Other Objectives

Table 17: Evaluation results with various loss objectives on linear model, showing (a) AUC across 15 datasets for classification and (b) RMSE across 7 datasets for regression. Best performances are bolded.

(a) Classification (AUC)

| Dataset | Supervised Contrastive Learning | CLIP | Reconstruction | Cross-entropy |
|---|---|---|---|---|
| Adult | 91.32±0.12 | 91.29±0.12 | **91.36±0.13** | 91.26±0.13 |
| Balance-scale | 99.51±0.33 | 99.51±0.22 | **99.65±0.27** | 99.57±0.32 |
| Bank | 92.08±0.18 | 92.06±0.15 | 91.95±0.28 | **92.19±0.23** |
| Blood | 74.85±2.89 | 74.54±3.17 | 74.78±2.56 | **74.96±2.98** |
| Car | 99.73±0.18 | 99.74±0.17 | **99.79±0.18** | 99.77±0.11 |
| Communities | 85.25±0.67 | **85.53±0.42** | 85.33±0.64 | 85.33±0.66 |
| Credit-g | **78.38±4.85** | 78.20±5.74 | 78.07±5.23 | 78.31±5.36 |
| Diabetes | **82.56±5.12** | 81.60±5.45 | 82.33±5.27 | 81.59±5.47 |
| Eucalyptus | 91.34±0.99 | 91.45±1.20 | **91.47±1.03** | 91.42±0.96 |
| Heart | **93.45±1.60** | 93.44±1.45 | 93.40±1.47 | 93.40±1.64 |
| Junglechess | **93.43±0.31** | 92.70±0.08 | 93.08±0.42 | 93.05±0.29 |
| Myocardial | **63.64±3.08** | 62.13±3.40 | 60.66±3.00 | 61.26±2.60 |
| Sequence-type | 96.44±1.01 | **96.63±0.77** | 96.41±1.05 | 96.31±0.78 |
| Tic-tac-toe | **99.52±0.51** | 99.49±0.32 | 99.47±0.25 | 99.39±0.42 |
| Vehicle | **96.22±0.28** | 96.12±0.57 | 96.15±0.48 | 96.05±0.58 |

(b) Regression (RMSE)

| Dataset | Supervised Contrastive Learning | CLIP | Reconstruction | Cross-entropy |
|---|---|---|---|---|
| Bike | **111.46±1.72** | 112.66±2.16 | 112.56±2.08 | 112.87±2.16 |
| Crab | 2.13±0.03 | **2.12±0.02** | **2.12±0.03** | **2.12±0.02** |
| Forest-fires | 83.19±22.47 | 87.95±24.07 | **82.04±22.03** | 82.45±22.59 |
| Housing | 56069.83±406.99 | **55967.38±301.06** | 56048.85±31.92 | 56381.03±248.59 |
| Insurance | 4578.14±149.83 | 4615.23±201.58 | 4644.91±122.84 | **4572.54±159.84** |
| Solution-mix | **0.02±0.01** | **0.02±0.00** | **0.02±0.00** | **0.02±0.00** |
| Wine | **0.67±0.00** | 0.68±0.00 | 0.68±0.00 | **0.67±0.00** |

## K.8 Experiments with Additional Datasets & Baselines

### K.8.1 Comparison on OpenML-CC18 datasets

Table 18: Evaluation results of self-supervised models on (a) linear model, (b) non-parametric classifier with 3 clusters, and (c) non-parametric classifier with 5 clusters, showing AUC across 19 OpenML-CC18 datasets. Additional datasets not included in the main paper are listed at the bottom of the table. Best performances are bolded, and ourframework's performances, when second-best, are underlined.

(a) Linear Model

| Dataset | Raw Data | AutoEncoder | SimSiam | SCARF | STAB | STUNT | LFR | FeatLLM | MET | Ours |
|---|---|---|---|---|---|---|---|---|---|---|
| Adult | 90.75±0.17 | 91.07±0.20 | 89.01±0.24 | 90.90±0.17 | 90.55±0.24 | 91.06±0.20 | 91.29±0.19 | 88.97±0.49 | 89.98±0.42 | **91.32±0.12** |
| Balance-scale | 97.24±1.11 | 99.58±0.37 | 99.37±0.46 | 99.44±0.39 | 97.66±1.46 | 93.10±2.50 | 99.28±0.39 | **100±0.00** | 78.81±1.93 | 99.51±0.33 |
| Bank | 90.48±0.18 | 91.14±0.07 | 87.32±0.16 | 91.73±0.04 | 90.06±0.24 | 91.10±0.38 | 91.65±0.17 | 88.21±0.17 | 90.96±0.16 | **92.08±0.18** |
| Blood | 75.15±3.21 | 74.98±3.52 | 75.18±4.42 | 73.92±4.04 | 74.75±3.24 | 74.39±4.83 | 73.88±3.11 | 66.65±1.07 | **75.87±1.61** | 74.85±2.89 |
| Car | 98.95±0.30 | 99.60±0.23 | 97.95±0.42 | 99.50±0.31 | 99.25±0.43 | 97.96±0.28 | **99.91±0.04** | 99.9±0.04 | 98.43±0.05 | 99.73±0.18 |
| Credit-g | 77.89±6.44 | 77.60±5.26 | 77.69±6.27 | 77.12±5.46 | 77.26±3.18 | 75.94±4.51 | 75.04±6.32 | 77.4±1.57 | 75.45±1.77 | **78.38±4.85** |
| Diabetes | 83.07±4.74 | 81.64±6.16 | 82.73±5.58 | 81.96±5.97 | 80.38±5.22 | 82.21±3.04 | 81.43±7.38 | 83.43±1.94 | **86.15±2.02** | 82.56±5.12 |
| Eucalyptus | **91.64±1.10** | 90.85±1.33 | 90.50±1.80 | 90.44±1.23 | 89.66±1.97 | 85.61±1.51 | 89.85±0.64 | 90.83±0.02 | 85.46±1.10 | 91.34±0.99 |
| Junglechess | 80.61±0.33 | 89.89±0.49 | 86.92±0.70 | 88.45±0.70 | 92.10±0.47 | 91.62±0.44 | 92.93±0.42 | 90.6±0.73 | 85.46±0.19 | **93.43±0.31** |
| Tic-tac-toe | 99.31±0.60 | 99.84±0.08 | 98.28±1.35 | 99.00±0.67 | 95.93±1.87 | 94.07±3.14 | 99.80±0.15 | **100±0.00** | 95.80±2.17 | 99.52±0.51 |
| Vehicle | 94.82±0.50 | 96.16±0.83 | 92.37±1.39 | 96.02±0.52 | 95.32±0.49 | 93.55±1.07 | **96.32±0.38** | 89.2±0.25 | 92.11±1.12 | 96.22±0.28 |
| Authorship | **100.00±0.00** | 99.90±0.04 | 99.90±0.08 | 99.78±0.34 | 99.98±0.02 | 99.99±0.01 | 99.99±0.01 | 99.36±0.35 | 99.70±0.21 | **100.00±0.00** |
| Breast-w | 99.55±0.44 | 99.43±0.41 | 99.42±0.49 | 99.12±1.21 | **99.60±0.38** | 98.16±1.27 | 98.84±0.56 | 98.60±0.48 | 99.29±0.24 | 99.45±0.38 |
| Cmc | 69.52±2.86 | 72.56±3.81 | 63.74±3.41 | 63.37±3.32 | 73.03±3.10 | 66.22±3.37 | 65.66±3.15 | **74.27±0.47** | 70.58±1.20 | 72.90±1.75 |
| Cylinder-bands | 84.53±3.98 | 72.73±3.27 | 49.64±2.44 | 74.87±4.81 | 81.03±1.64 | 66.96±4.12 | **85.92±4.05** | 76.75±1.82 | 60.03±1.36 | 83.36±3.24 |
| Dmft | 59.35±1.12 | 59.57±1.08 | 59.86±1.44 | 59.86±1.64 | 60.67±1.26 | 55.67±0.90 | 53.45±2.49 | 56.59±0.40 | **76.95±1.80** | 59.52±0.80 |
| Ilpd | **77.73±3.34** | 77.05±3.51 | 73.79±1.10 | 76.39±4.14 | 77.50±3.31 | 74.23±2.67 | 76.12±4.66 | 73.35±1.17 | 58.42±2.84 | 77.53±2.42 |
| Optdigits | 99.92±0.02 | 99.60±0.12 | 84.60±0.26 | 97.64±0.36 | 99.92±0.03 | 94.36±0.42 | 99.89±0.12 | 99.36±0.01 | 99.69±0.04 | **99.97±0.01** |
| Pc1 | 82.45±3.53 | 79.92±5.88 | 74.21±7.02 | 76.48±7.59 | 84.47±1.49 | 78.07±4.03 | 76.88±17.95 | 83.66±2.52 | 86.88±1.09 | **87.54±2.63** |

(b) Non-parametric Classifier with 3 Clusters

| Dataset | Raw Data | AutoEncoder | SimSiam | SCARF | STAB | STUNT | LFR | FeatLLM | MET | Ours |
|---|---|---|---|---|---|---|---|---|---|---|
| Adult | **82.31±0.18** | 81.53±0.16 | 80.51±0.41 | 81.90±0.09 | 82.21±0.24 | 82.20±0.30 | 82.28±0.29 | 81.63±0.17 | 80.81±0.29 | 81.90±0.46 |
| Balance-scale | 79.47±2.44 | 78.40±1.60 | 82.67±1.85 | 78.67±0.46 | 79.73±3.23 | 79.20±1.60 | 78.13±2.44 | **98.80±1.70** | 65.87±3.03 | 79.73±0.46 |
| Bank | 89.09±0.14 | 89.19±0.03 | 88.41±0.25 | 89.15±0.04 | 89.11±0.04 | 88.96±0.11 | 89.16±0.11 | 88.34±0.22 | 88.34±0.22 | **89.36±0.26** |
| Blood | 72.44±4.44 | 71.33±3.46 | 71.33±4.67 | 69.78±5.00 | 71.33±4.67 | **74.67±3.71** | 72.00±4.16 | 72.67±1.89 | 70.44±2.78 | 74.33±3.06 |
| Car | 87.09±2.09 | 86.42±2.37 | 77.17±1.16 | 79.00±0.73 | 82.01±1.59 | 82.85±1.97 | 79.77±2.08 | 81.36±0.61 | 88.92±6.46 | **89.40±2.73** |
| Credit-g | 73.00±1.50 | **73.33±0.76** | 69.00±4.44 | 71.83±3.33 | 72.00±3.04 | 71.67±4.65 | 71.50±1.00 | 73.00±0.00 | 70.50±1.32 | 71.83±1.76 |
| Diabetes | 73.16±4.96 | 72.94±6.57 | 69.91±7.30 | 70.78±3.95 | **74.24±1.35** | 74.03±4.06 | 72.51±4.12 | 74.03±2.75 | 71.43±1.72 | 72.73±4.90 |
| Eucalyptus | 59.23±3.19 | 53.60±3.96 | 57.88±2.73 | 52.03±4.05 | 56.08±4.22 | 52.25±2.06 | 58.23±2.06 | **61.82±1.43** | 58.28±5.87 | 60.59±5.12 |
| Junglechess | 75.08±0.54 | 74.35±0.27 | **77.40±0.14** | 72.34±0.22 | 74.84±0.64 | 73.65±0.47 | 73.87±0.38 | 75.21±1.44 | 72.31±0.65 | 75.35±0.52 |
| Tic-tac-toe | 91.15±0.52 | 82.29±0.90 | 85.07±2.41 | 84.90±1.56 | 96.70±1.59 | 77.95±2.46 | | **99.48±0.00** | 86.11±4.37 | 93.92±1.59 |
| Vehicle | 69.80±2.96 | **75.10±3.55** | 59.80±2.96 | 68.82±5.23 | 68.80±1.80 | 67.59±3.67 | 74.31±4.34 | 66.18±5.41 | 63.51±4.44 | 74.51±2.65 |
| Authorship | 99.11±0.34 | 94.87±3.36 | 38.66±4.03 | 87.77±6.83 | 99.21±0.34 | **99.61±0.34** | 90.73±5.93 | 89.35±0.84 | 93.45±0.00 | 98.82±0.59 |
| Breast-w | **95.95±2.51** | 94.29±0.00 | 89.52±2.18 | 95.95±1.49 | 95.95±1.49 | 95.95±1.65 | 94.52±1.49 | 95.71±2.02 | 96.19±0.41 | 94.76±1.49 |
| Cmc | 45.42±1.55 | **47.68±2.82** | 40.34±2.56 | 44.29±1.37 | 45.20±2.07 | 45.54±1.04 | 43.84±2.26 | 45.25±3.12 | 45.42±2.37 | 45.42±3.11 |
| Cylinder-bands | 64.20±4.57 | 68.21±2.83 | 50.93±0.93 | 72.84±1.41 | **78.09±5.43** | 60.49±2.98 | 77.47±3.74 | 71.30±1.31 | 74.38±4.57 | 74.69±4.18 |
| Dmft | 17.71±3.08 | 18.33±1.30 | 18.33±0.36 | 15.83±1.30 | 16.67±1.30 | **19.38±2.17** | 17.50±3.12 | 16.56±0.44 | 15.72±2.74 | 17.71±2.01 |
| Ilpd | 68.66±0.49 | 64.39±4.04 | 60.97±6.06 | 69.80±1.31 | 67.24±4.93 | **70.09±1.48** | 67.52±3.08 | 70.09±2.42 | 68.38±5.13 | 65.81±3.08 |
| Optdigits | **97.84±0.29** | 95.58±0.62 | 32.71±1.91 | 77.79±0.27 | 97.51±0.18 | 79.12±0.81 | 97.75±0.52 | 88.48±0.44 | 90.51±2.06 | 97.12±0.67 |
| Pc1 | 93.54±1.04 | 93.54±0.69 | 90.09±0.78 | 92.94±1.82 | 93.09±0.52 | 93.54±0.94 | 92.49±0.69 | 93.47±2.23 | **93.99±1.71** | 93.69±0.90 |

(c) Non-parametric Classifier with 5 Clusters

| Dataset | Raw Data | AutoEncoder | SimSiam | SCARF | STAB | STUNT | LFR | FeatLLM | MET | Ours |
|---|---|---|---|---|---|---|---|---|---|---|
| Adult | 83.17±0.19 | 82.48±0.12 | 81.47±0.31 | 82.60±0.11 | 83.17±0.11 | 82.93±0.35 | 83.15±0.37 | 81.65±0.22 | 81.53±0.34 | **83.22±0.32** |
| Balance-scale | 82.40±2.88 | 82.40±2.12 | 86.67±0.46 | 83.20±2.40 | 81.60±3.20 | 85.07±2.81 | 81.33±3.33 | **98.00±2.83** | 69.87±5.90 | 81.33±0.92 |
| Bank | 89.40±0.32 | 89.48±0.14 | 88.85±0.22 | 89.13±0.16 | **89.54±0.04** | 89.37±0.26 | 89.47±0.27 | 88.08±0.10 | 89.05±0.17 | 89.25±0.06 |
| Blood | **74.67±4.16** | 74.44±4.07 | 73.56±4.73 | 74.44±2.78 | **74.67±4.91** | 74.22±4.34 | 72.89±3.67 | 73.67±4.24 | 71.78±2.69 | 73.78±4.07 |
| Car | 89.69±0.83 | 84.90±1.01 | 78.13±1.77 | 85.07±1.64 | 88.54±2.53 | 88.54±2.67 | 84.78±1.92 | 83.09±0.20 | 91.33±4.26 | **93.77±2.29** |
| Credit-g | 72.33±2.57 | **74.83±0.29** | 71.17±4.54 | 73.50±3.61 | 71.83±2.93 | 71.33±1.15 | 71.67±1.53 | 70.00±2.12 | 70.50±3.61 | 73.17±3.88 |
| Diabetes | 73.38±3.25 | 72.51±4.32 | 73.38±5.15 | 73.16±2.46 | 72.94±3.33 | **74.03±3.62** | 73.38±3.90 | 73.70±4.13 | 73.16±3.20 | 71.65±5.52 |
| Eucalyptus | 59.46±4.87 | 54.95±3.96 | 58.33±2.56 | 53.60±3.47 | 54.28±5.46 | 52.70±3.10 | 58.33±3.96 | 60.14±2.87 | 56.01±7.27 | **63.16±3.10** |
| Junglechess | 75.20±0.42 | 75.28±0.40 | **78.93±0.57** | 74.04±0.54 | 76.09±0.61 | 75.57±0.33 | 75.23±0.52 | 75.97±1.62 | 72.99±0.51 | 75.80±0.43 |
| Tic-tac-toe | 94.10±0.80 | 84.38±1.80 | 87.33±1.97 | 77.26±2.46 | 90.45±2.87 | 97.74±0.60 | 82.12±1.08 | **100.00±0.00** | 90.28±4.37 | 95.31±1.88 |
| Vehicle | 72.75±1.70 | 75.49±0.90 | 60.39±0.34 | 70.39±3.59 | 72.75±0.34 | 71.57±2.23 | 74.71±3.11 | 67.65±5.82 | 62.13±5.33 | **76.47±3.53** |
| Authorship | 99.11±0.34 | 97.24±1.23 | 40.04±1.81 | 88.17±5.64 | 99.41±0.59 | **99.61±0.34** | 87.97±7.88 | 88.76±0.84 | 93.45±1.03 | 98.62±0.34 |
| Breast-w | 95.48±0.41 | 94.76±0.41 | 89.05±1.65 | 95.71±1.24 | **96.43±1.89** | 96.19±2.06 | 95.71±0.71 | 95.71±1.01 | 96.19±0.41 | 94.29±1.24 |
| Cmc | 45.08±1.55 | 47.34±1.99 | 40.56±1.60 | 46.10±3.44 | 45.42±1.17 | 45.31±1.09 | 45.65±3.33 | 46.10±0.48 | 47.12±1.22 | **48.02±1.99** |
| Cylinder-bands | 62.65±2.14 | 68.21±4.66 | 52.78±3.34 | 70.99±2.33 | **79.32±2.98** | 62.35±4.28 | 75.00±3.34 | 68.06±1.96 | 71.91±1.93 | 71.91±3.25 |
| Dmft | 15.62±3.25 | **19.38±2.25** | 18.75±2.17 | 17.08±3.08 | 18.12±3.48 | 19.17±2.01 | 18.12±4.38 | 16.56±2.21 | 17.19±0.96 | 17.71±1.91 |
| Ilpd | 66.95±5.49 | 65.53±4.93 | 62.96±5.56 | 69.23±2.26 | 68.09±2.15 | **72.36±3.00** | 68.09±2.75 | 71.37±5.44 | 67.52±0.00 | 68.95±3.45 |
| Optdigits | **97.78±0.18** | 95.02±1.00 | 33.90±1.11 | 79.12±0.59 | 97.48±0.40 | 79.18±0.50 | 97.66±0.40 | 88.97±0.13 | 90.48±1.55 | 97.24±0.73 |
| Pc1 | 92.79±0.78 | 94.29±0.52 | 91.14±1.38 | 93.24±1.35 | 93.24±0.78 | 92.94±0.26 | 93.54±0.69 | 94.37±0.32 | **94.59±1.56** | 93.39±0.26 |

### K.8.2 Comparison with additional baselines

Table 19: Evaluation results of self-supervised models on linear model, showing (a) AUC across 23 datasets for classification and (b) RMSE across 7 datasets for regression. Best performances are bolded, and ourframework's performances, when second-best, are underlined

#### (a) Classification (AUC)

| Dataset | PCA | t-SNE | XGBoost | Random Forest | LightGBM | EBM | TabPFN | TP-BERTa | Ours |
|---|---|---|---|---|---|---|---|---|---|
| Adult | 89.14±0.16 | 83.60±0.21 | 92.95±0.23 | 90.35±0.20 | 92.99±0.15 | **93.24±0.15** | 88.73±0.54 | 92.96±0.02 | 91.32±0.12 |
| Balance-scale | 97.24±0.91 | 71.20±5.81 | 91.87±0.27 | 81.80±1.72 | 91.80±0.74 | 99.97±0.04 | **99.98±0.03** | 93.60±2.85 | 99.51±0.33 |
| Bank | 86.56±0.43 | 66.14±0.80 | 93.18±0.21 | 92.70±0.27 | 93.57±0.11 | 93.22±0.13 | 89.69±0.34 | **96.64±0.10** | 92.08±0.18 |
| Blood | **75.15±2.62** | 72.18±2.91 | 69.66±3.76 | 66.94±4.95 | 70.71±4.19 | 72.92±3.45 | 74.57±4.54 | 68.56±6.36 | 74.85±2.89 |
| Car | 90.85±2.00 | 67.79±9.52 | 99.92±0.11 | 99.65±0.21 | **99.97±0.04** | 99.21±0.25 | 99.70±0.05 | 97.88±1.57 | 99.73±0.18 |
| Communities | 84.74±0.52 | 80.04±1.34 | **85.75±0.22** | 85.44±0.37 | 85.63±0.45 | 84.90±0.33 | 85.01±0.18 | 65.66±0.35 | 85.25±0.67 |
| Credit-g | 75.26±7.13 | 55.21±1.39 | **79.41±5.61** | 78.67±4.05 | 78.40±3.64 | 77.27±4.76 | 77.25±6.60 | 69.55±2.30 | 78.38±4.85 |
| Diabetes | 83.07±3.87 | 64.83±0.64 | 79.44±3.92 | 82.10±3.09 | 80.48±2.40 | **83.39±2.69** | 81.75±4.37 | 73.58±1.67 | 82.56±5.12 |
| Eucalyptus | 88.16±1.33 | 59.85±1.34 | 90.76±0.62 | 90.82±0.78 | 90.17±0.24 | 91.16±0.41 | **93.23±0.57** | 60.81±2.53 | 91.34±0.99 |
| Heart | 92.59±1.48 | 91.04±1.91 | 92.89±2.13 | **93.96±1.11** | 93.43±1.62 | 93.17±2.16 | 93.29±1.56 | 92.11±0.98 | 93.45±1.60 |
| Junglechess | 80.61±0.27 | 59.31±1.30 | **97.73±0.10** | 93.75±0.21 | 97.38±0.12 | 86.89±0.23 | 91.75±0.58 | 95.04±1.58 | 93.43±0.31 |
| Myocardial | 63.15±5.48 | 51.52±3.67 | 62.03±3.02 | 66.06±1.61 | 63.75±0.88 | 63.44±3.09 | **70.14±2.28** | 63.28±0.30 | 63.64±3.08 |
| Sequence-type | 92.22±1.62 | 69.98±3.14 | 95.87±2.32 | 96.23±2.01 | 96.70±2.18 | 95.29±2.51 | **99.17±0.34** | 76.00±4.90 | 96.44±1.01 |
| Tic-tac-toe | 74.88±3.95 | 46.55±6.83 | 99.92±0.12 | **99.94±0.05** | 99.92±0.10 | 99.88±0.02 | 99.90±0.04 | 98.74±1.25 | 99.52±0.51 |
| Vehicle | 86.48±0.79 | 66.70±0.63 | 93.10±0.99 | 93.11±0.32 | 93.00±0.47 | 92.67±0.33 | 96.18±0.61 | 64.90±2.17 | **96.22±0.28** |
| Authorship | **100.00±0.00** | 99.90±0.08 | 99.94±0.03 | 99.97±0.01 | 99.98±0.01 | 99.98±0.01 | **100.00±0.00** | 90.53±1.67 | **100.00±0.00** |
| Breast-w | 99.55±0.45 | 98.89±0.54 | 98.87±0.65 | 99.19±0.62 | 98.68±0.56 | 99.18±0.55 | **99.60±0.42** | 97.31±1.59 | 99.45±0.38 |
| Cmc | 69.51±2.39 | 60.98±1.67 | 69.42±2.47 | 68.74±1.53 | 70.11±1.99 | 73.21±3.45 | **74.14±2.54** | 55.25±0.73 | 72.90±1.75 |
| Cylinder-bands | 71.17±0.99 | 55.11±4.47 | **92.12±3.63** | 90.53±3.58 | 92.02±1.72 | 90.66±3.20 | 83.08±0.99 | 81.88±1.66 | 83.36±3.24 |
| Dmft | 58.15±0.87 | 51.49±0.84 | 54.87±2.25 | 52.65±2.80 | 55.36±1.80 | 59.04±0.85 | 58.56±1.75 | 16.25±2.70 | **59.52±0.80** |
| Ilpd | 77.90±2.29 | 70.67±0.68 | 73.52±5.16 | 74.49±4.77 | 74.52±4.49 | 75.79±4.20 | **78.14±2.38** | 44.94±3.61 | 77.53±2.42 |
| Optdigits | 98.75±0.12 | 98.34±0.15 | 99.95±0.02 | 99.96±0.02 | 99.96±0.02 | 99.93±0.01 | 99.93±0.02 | 93.36±0.67 | **99.97±0.01** |
| Pc1 | 80.96±4.69 | 73.04±8.14 | 86.73±6.63 | 85.85±1.96 | 88.09±2.70 | 84.71±3.33 | **91.69±4.76** | 57.33±8.26 | 87.54±2.63 |

#### (b) Regression (RMSE)

| Dataset | PCA | t-SNE | XGBoost | Random Forest | LightGBM |
|---|---|---|---|---|---|
| Bike | 142.63±1.35 | 178.41±0.37 | **41.30±1.43** | 44.11±1.39 | 42.90±1.45 |
| Crab | 2.25±0.04 | 2.70±0.07 | 2.33±0.06 | 2.23±0.01 | 2.23±0.02 |
| Forest-fires | **74.16±29.94** | 74.52±30.67 | 99.70±41.20 | 80.72±28.65 | 77.95±26.25 |
| Housing | 70262.25±417.53 | 101805.39±352.75 | **47142.65±721.83** | 48536.76±310.72 | 47589.89±596.00 |
| Insurance | 6184.51±202.65 | 12061.28±304.16 | 5095.32±116.84 | 4655.27±187.93 | 4626.40±170.12 |
| Solution-mix | 0.07±0.00 | 0.17±0.02 | 0.08±0.00 | 0.08±0.01 | 0.07±0.00 |
| Wine | 0.75±0.01 | 0.83±0.01 | 0.71±0.00 | **0.60±0.01** | 0.64±0.00 |

| Dataset | EBM | TabPFN | TP-Berta | Ours |
|---|---|---|---|---|
| Bike | 54.53±1.43 | - | 42.17±0.40 | 111.46±1.72 |
| Crab | 2.18±0.03 | - | 2.19±0.03 | **2.13±0.03** |
| Forest-fires | 74.36±30.47 | - | 108.69±0.12 | 83.19±22.47 |
| Housing | 47782.57±1172.50 | - | 49205.93±340.66 | 56069.83±406.99 |
| Insurance | **4177.76±179.95** | - | 4519.31±13.08 | 4578.14±149.83 |
| Solution-mix | 0.05±0.00 | - | 0.12±0.01 | **0.02±0.01** |
| Wine | 0.68±0.00 | - | 0.71±0.00 | 0.67±0.00 |

### K.8.3 Comparison with Other Classifiers

Table 20: Evaluation results of various classifiers on raw data and on our model's embeddings. Bolded values indicates that our model outperformed the raw data.

| Dataset | Linear Model Raw Data | Linear Model Ours | XGBoost Raw Data | XGBoost Ours | EBM Raw Data | EBM Ours | MLP Raw Data | MLP Ours | TabNet Raw Data | TabNet Ours |
|---|---|---|---|---|---|---|---|---|---|---|
| Adult | 90.75±0.17 | **91.32±0.12** | 92.95±0.23 | 89.69±0.07 | 93.24±0.15 | 91.11±0.12 | 88.01±0.07 | **88.89±0.02** | 87.26±0.87 | **87.92±0.20** |
| Balance-scale | 97.24±1.11 | **99.51±0.33** | 91.87±0.27 | **96.86±2.03** | 99.97±0.04 | 95.24±2.16 | 99.91±0.08 | 99.90±0.05 | 95.21±3.49 | **96.43±1.06** |
| Bank | 90.48±0.18 | **92.08±0.18** | 93.18±0.21 | 90.90±0.40 | 93.22±0.13 | 92.14±0.28 | 90.09±0.28 | **90.73±0.13** | 89.17±1.02 | 88.86±0.50 |
| Blood | 75.15±3.21 | 74.85±2.89 | 69.66±3.76 | 67.41±4.67 | 72.92±3.45 | **74.69±3.22** | 73.82±2.90 | **73.95±3.36** | 70.55±4.43 | **71.95±3.65** |
| Car | 98.95±0.30 | **99.73±0.18** | 99.92±0.11 | 99.39±0.13 | 99.21±0.25 | **99.82±0.09** | 99.99±0.01 | **100.00±0.00** | 100.00±0.00 | 99.93±0.05 |
| Communities | 84.31±1.23 | **85.25±0.67** | 85.75±0.22 | 84.07±0.62 | 84.90±0.33 | **85.54±0.86** | 81.97±1.18 | **82.14±1.38** | 80.51±1.62 | 79.01±0.73 |
| Credit-g | 77.89±6.44 | **78.38±4.85** | 79.41±5.61 | 75.95±5.37 | 77.27±4.76 | **78.15±6.36** | 75.90±4.29 | **77.41±4.45** | 65.31±6.96 | **66.40±4.77** |
| Diabetes | 83.07±4.74 | 82.56±5.12 | 79.44±3.92 | **80.95±5.96** | 83.39±2.69 | 81.64±5.17 | 80.14±6.66 | 75.28±9.50 | 74.99±2.90 | **76.22±3.79** |
| Eucalyptus | 91.64±1.10 | 91.34±0.99 | 90.76±0.62 | 89.08±1.35 | 91.16±0.41 | 91.14±1.07 | 92.48±0.53 | **92.91±0.44** | 87.15±1.25 | **88.95±1.97** |
| Heart | 93.10±2.12 | **93.45±1.60** | 92.89±2.13 | 92.65±0.99 | 93.17±2.16 | 93.09±1.63 | 92.64±1.62 | 91.26±2.05 | 90.46±0.91 | 89.61±0.79 |
| Junglechess | 80.61±0.33 | **93.43±0.31** | 97.73±0.10 | 96.61±0.12 | 86.89±0.23 | **95.75±0.20** | 98.04±0.08 | **98.90±0.04** | 98.16±0.61 | **98.95±0.18** |
| Myocardial | 61.20±5.13 | **63.64±3.08** | 62.03±3.02 | **63.08±4.70** | 63.44±3.09 | 61.33±2.94 | 56.49±2.20 | **62.44±1.04** | 60.12±2.32 | 55.62±6.88 |
| Sequence-type | 92.11±2.03 | **96.44±1.01** | 95.87±2.32 | **98.16±1.27** | 95.29±2.51 | **98.35±1.11** | 96.88±1.26 | **97.18±0.52** | 94.45±3.02 | 89.31±5.26 |
| Tic-tac-toe | 99.31±0.60 | **99.52±0.51** | 99.92±0.12 | 99.29±0.21 | 99.88±0.02 | 99.82±0.20 | 99.96±0.04 | **99.97±0.02** | 99.30±0.93 | **99.71±0.39** |
| Vehicle | 94.82±0.50 | **96.22±0.28** | 93.10±0.99 | **94.58±0.55** | 92.67±0.33 | **94.80±0.24** | 96.25±0.26 | **96.40±0.38** | 95.12±0.61 | 93.82±0.74 |
| Authorship | 100.00±0.00 | 100.00±0.00 | 99.94±0.03 | 99.92±0.00 | 99.98±0.01 | **99.99±0.01** | 99.99±0.01 | 99.99±0.01 | 99.30±0.79 | **99.62±0.24** |
| Breast-w | 99.55±0.44 | 99.45±0.38 | 98.87±0.65 | **99.27±0.60** | 99.18±0.55 | 98.97±0.91 | 99.10±0.64 | **99.11±0.53** | 97.83±0.97 | 97.04±2.06 |
| Cmc | 69.52±2.86 | **72.90±1.75** | 69.42±2.47 | 68.96±2.21 | 73.21±3.45 | **74.46±2.85** | 72.49±2.21 | 70.70±0.97 | 66.94±3.18 | **69.23±2.96** |
| Cylinder-bands | 84.53±3.98 | 83.36±3.24 | 92.12±3.63 | 81.03±4.83 | 90.66±3.20 | 83.82±3.01 | 87.01±2.81 | 85.95±3.35 | 78.50±3.73 | 76.54±2.44 |
| Dmft | 59.35±1.12 | **59.52±0.80** | 54.87±2.25 | 53.21±1.86 | 59.04±0.85 | **60.42±1.24** | 59.09±0.78 | 58.56±0.63 | 55.13±2.17 | 53.11±2.31 |
| Ilpd | 77.73±3.34 | 77.53±2.42 | 73.52±5.16 | **78.55±0.94** | 75.79±4.20 | **78.83±2.46** | 76.59±2.83 | 75.69±2.97 | 70.93±5.71 | **75.80±4.73** |
| Optdigits | 99.92±0.02 | **99.97±0.01** | 99.95±0.02 | 99.90±0.04 | 99.93±0.01 | **99.97±0.01** | 99.98±0.00 | 99.94±0.04 | 99.87±0.02 | 99.65±0.11 |
| Pc1 | 82.45±3.53 | **87.54±2.63** | 86.73±6.63 | 90.21±2.24 | 84.71±3.33 | 90.54±3.55 | 84.48±4.09 | 84.19±7.58 | 77.62±7.43 | **82.07±6.28** |

### K.9 Impact of the Number of LLM Trials

Table 21: Evaluation results with various numbers of trials on linear model, showing (a) AUC across 15 datasets for classification and (b) RMSE across 7 datasets for regression. Best performances are bolded.

(a) Classification (AUC)

| Number of Trials | 5 | 10 | 20 | 30 | 40 |
|---|---|---|---|---|---|
| Adult | **91.32±0.14** | 91.31±0.09 | **91.32±0.08** | 91.29±0.08 | **91.32±0.12** |
| Balance-scale | **99.67±0.20** | 99.66±0.22 | 99.58±0.30 | 99.53±0.33 | 99.51±0.33 |
| Bank | 91.98±0.15 | 92.06±0.38 | 91.91±0.29 | 91.96±0.22 | **92.08±0.18** |
| Blood | 74.81±2.84 | 74.94±2.99 | **75.14±2.98** | 74.84±2.90 | 74.85±2.89 |
| Car | 99.56±0.24 | 99.57±0.24 | 99.73±0.13 | **99.75±0.14** | 99.73±0.18 |
| Communities | **85.27±0.52** | 85.13±0.64 | 85.01±0.95 | 85.18±0.79 | 85.25±0.67 |
| Credit-g | 77.55±4.74 | 78.21±5.70 | **78.77±4.80** | 78.60±5.35 | 78.38±4.85 |
| Diabetes | 82.07±5.48 | 81.94±5.38 | 81.76±5.16 | 81.71±5.54 | **82.56±5.12** |
| Eucalyptus | 91.12±0.99 | 91.56±0.68 | **91.71±0.71** | 91.32±1.01 | 91.34±0.99 |
| Heart | 93.69±1.37 | **93.98±1.57** | 93.52±1.29 | 93.55±1.27 | 93.45±1.60 |
| Junglechess | 93.45±0.28 | **93.70±0.40** | 93.48±0.32 | 93.60±0.39 | 93.43±0.31 |
| Myocardial | 61.15±3.08 | 61.67±2.08 | **64.05±2.48** | 64.03±2.35 | 63.64±3.08 |
| Sequence-type | 96.50±0.73 | 96.30±1.03 | **96.66±1.09** | 96.50±0.71 | 96.44±1.01 |
| Tic-tac-toe | **99.73±0.18** | 99.63±0.22 | 99.52±0.54 | 99.34±0.39 | 99.52±0.51 |
| Vehicle | 96.17±0.44 | 95.96±0.37 | 96.04±0.29 | 96.07±0.49 | **96.22±0.28** |

(b) Regression (RMSE)

| Number of Trials | 5 | 10 | 20 | 30 | 40 |
|---|---|---|---|---|---|
| Bike | 111.39±2.14 | 111.60±2.09 | **111.10±1.76** | 112.38±2.43 | 111.46±1.72 |
| Crab | 2.13±0.02 | 2.13±0.02 | 2.13±0.03 | **2.12±0.02** | 2.13±0.03 |
| Forest-fires | 82.46±22.67 | 81.84±22.78 | **81.56±23.50** | 81.75±21.90 | 83.19±22.47 |
| Housing | **55959.59±258.33** | 56001.28±136.26 | 56124.43±103.62 | 56134.17±310.96 | 56069.83±406.99 |
| Insurance | 4612.62±200.13 | 4618.29±179.43 | **4576.85±156.01** | 4598.24±186.34 | 4578.14±149.83 |
| Solution-mix | **0.02±0.00** | **0.02±0.00** | **0.02±0.00** | **0.02±0.00** | **0.02±0.01** |
| Wine | 0.68±0.00 | 0.68±0.01 | 0.68±0.00 | 0.68±0.00 | **0.67±0.00** |

