# OpenReview forum: "LLM-Guided Self-Supervised Tabular Learning With Task-Specific Pre-text Tasks"
_TMLR — Accepted by TMLR_

### Review · Reviewer_zHjZ · 2025-02-04

**Summary Of Contributions:**

The paper presents TST-LLM, a novel framework for tabular self-supervised representation learning. By defining pre-text tasks in a task-specific manner using natural language descriptions, TST-LLM claims to identify and combine relevant features, boosting performance on downstream tasks compared to strong baseline methods.

**Audience:**

Yes

**Claims And Evidence:**

No

**Requested Changes:**

1. An appropriate choice of baseline comparison models is challlenging for this paper, because the different components of the model are operating with different degrees of awareness of the data. While the authors present this as an unsupervised/self-supervised method, and the LLM Feature Discovery and Self-supervised Pretraining components do not rely on labels, the authors use all of the training data labels to fit the final (linear or non-parametric) classifier in the main paper experiments.

However, the strongest baseline methods the authors have chosen to compare to include;

STUNT: Designed for few-shot learning scenarios

FeatLLM: Focused on few-shot tabular learning

LFR: Development was motivated by scenarios with limited or noisy labels

SCARF: Development was motivated by scenarios with limited or noisy labels

Essentially, there is a mismatch between the authors' experimental setting (which ultimately requires full labeled datasets and limits the choice of classifier architecture) and several of their chosen baselines (few-shot/limited data methods, many of which rely on particular classifier architectures). This could potentially paint an overly favorable picture of their method's performance.

Therefore, I would like to see comparisons to general-purpose tabular learning methods, such as XGBoost, CatBoost, TabPFN, XTFormer, etc, with 30 iterations of HPO (for those methods which require it) and parameter search ranges sampled from recent literature (https://arxiv.org/abs/2305.02997), added to the main paper. The authors should feel free to use either linear or non-parametric classifier for these experiments.

In addition, I would like to see comparisons to strong self-supervised tabular baselines which are designed to work in the large-data regime without label noise, such as the recent TP-Berta (https://arxiv.org/html/2403.01841v2).

2. The authors have also chosen a non-standard benchmark suite; "a total of 22 datasets to ensure a diverse range of downstream tasks in terms of size and complexity". This choice is insufficiently motivated by the paper. Non-standard benchmark suites make it unnecessarily difficult for readers and reviewers to understand the authors' overall contribution in the context of prior work, most of which, in this case, has consistently used OpenML-CC18. Therefore, I would ask the authors to redo their experiments using a standard benchmark, ideally OpenML-CC18, or at least a subset consistent with prior work to ensure consistency, including those requested in (1). This can be an appendix result, and the authors need not show SOTA performance.

Both (1) and (2) are critical to securing my recommendation for acceptance.

**Strengths And Weaknesses:**

STRENGTHS:

The work is well-documented; figures are useful and appropriately placed. The setting of tabular classification and regression, although extremely well-studied, remains highly relevant, and the attempt to integrate LLMs into the process is timely. The authors are to be commended for including many strong baseline methods; however, that commendation comes with a significant caveat (see req'd changes).

WEAKNESSES:

See below.

---

> ### Author Response · Authors · 2025-03-03
> **Response to Reviewer zHjZ**
>
> Thank you so much for your valuable feedback. We have carefully revised our paper based on your comments. If you have any further questions, please let us know!
>
> > 1. Comparison to general-purpose supervised tabular learning methods, and a strong self-supervised tabular baseline.
>
> As suggested, we have conducted additional comparisons between TST-LLM with linear model as the classifier and various ensemble approaches, such as XGBoost, Random Forest, LightGBM, and Explainable Boosting Machine, as well as some recent deep learning-based approaches, such as TabPFN and TP-BERTa. The results have been incorporated into Appendix F.2. Our findings indicate that representations learned by our model (with linear model as the classifier) generally outperform or are at least comparable to these methods.
>
> We would also like to clarify is that neither TabPFN nor TP-BERTa serve as self-supervised tabular baselines, as they are trained on a large amount of labeled synthetic/benchmark tabular datasets and transfer this knowledge to downstream tasks.
>
> > 2. Comparison over OpenML-CC18 dataset.
>
> We conducted benchmark dataset experiments by referring to previous works, FeatLLM [1] and TabLLM [2]. Upon review, we found that 11 of the datasets we used (e.g., Adult, Bank) are included in OpenML-CC18. As suggested, we expanded our evaluation by adding datasets from OpenML-CC18 and aggregated the results in Appendix F.1. Since our model requires task and feature descriptions, certain datasets could not be applied due to the absence of this meta-information. We conducted experiments on 8 additional datasets that included the necessary metadata. The new results show that our model has better performance on additional datasets. In addition to this, we now discuss limitations in Conclusion. Thank you for your insightful comment.
>
> [1] Large Language Models Can Automatically Engineer Features for Few-Shot Tabular Learning (ICML’24)
> [2] TabLLM: Few-shot Classification of Tabular Data with Large Language Models (AISTATS’23)

---

> > ### Comment · Reviewer_zHjZ · 2025-03-04
> >
> > I thank the authors for a concise and informative reply. The added experiments are certainly helpful, and I will factor them into my eventual decision on the paper.

---

### Review · Reviewer_D9b8 · 2025-02-10

**Summary Of Contributions:**

The paper introduces TST-LLM, a novel approach to representation learning that leverages features discovered by large language models (LLMs) without relying on ground-truth labels or label statistics. This method is designed to solve task-specific pre-text tasks through a two-step process. First, the target task’s textual description, feature meta-information, and text-serialized unlabeled data are used to construct prompts, which are then fed into an LLM to extract task-relevant features. These discovered features serve as pseudo ground-truth labels, defining a pre-text task that is optimized through supervised contrastive learning.

**Audience:**

Yes

**Claims And Evidence:**

Yes

**Requested Changes:**

See Strengths And Weaknesses.

**Strengths And Weaknesses:**

The introduction effectively presents the problem but lacks an overview of the document’s structure and concrete examples of applications within the experimental setting.  Overall, the paper is well-written and well-presented, with a clear methodology. However, incorporating a running example would further clarify the workflow for readers. The experimental section is thorough, covering various aspects, including ablation studies and comparisons with baselines and alternative methods. To strengthen the analysis, I recommend adding a baseline using raw data combined with traditional feature reduction techniques (e.g., PCA, t-SNE) and evaluating additional models such as decision trees and ensemble methods (Random Forest, XGBoost, LightGBM, Explainable Boosting Machine). This would help assess whether the augmented feature space benefits state-of-the-art models for tabular data. Lastly, the conclusion is quite minimal and does not discuss potential directions for future research, which would be a valuable addition to the paper.

---

> ### Author Response · Authors · 2025-03-03
> **Response to Reviewer D9b8**
>
> Thank you so much for your valuable feedback. We have carefully revised our paper based on your comments. If you have any further questions, please let us know!
>
> > 1. The introduction lacks an overview of the document’s structure and concrete examples of applications within the experimental setting. Incorporating a running example would further clarify the workflow for readers.
>
> Thank you for your suggestion. In the revised manuscript, we have included an example of a diabetes classification task in Introduction and enhanced Figure 2 to clarify the workflow of our model through a running example. Additionally, we have added Figure 1 to visually highlight the differences between our model and previous works. Finally, we have provided more detailed descriptions of the document’s structure at the end of Introduction.
>
>
> > 2. Adding a baseline using raw data combined with traditional feature reduction techniques and additional models such as decision trees and ensemble methods.
>
> As recommended, we have included results comparing our model with traditional feature reduction techniques such as PCA and t-SNE in Appendix F.2. Additionally, we combined various ensemble approaches, including XGBoost, Random Forest, LightGBM, and Explainable Boosting Machine, on top of our model’s trained embeddings to evaluate whether our model can benefit state-of-the-art models. These results have been added to Appendix F.3. Our findings indicate that representations learned by our model outperform traditional feature reduction techniques and enhance the performance of existing tabular machine learning models.
>
>
> > 3. The conclusion is quite minimal and does not discuss potential directions for future research, which would be a valuable addition to the paper.
>
> Thank you for pointing this out. We have shown that incorporating task meta-information and the prior knowledge of LLM is highly effective for representation learning across diverse domains. For future work, we suggest extending beyond self-supervised learning to semi-supervised learning by incorporating labeled samples. In cases where meta-information alone is insufficient to define relationships with target labels or where relying on prior knowledge is challenging for specific tasks, labeled samples can serve as a complement. We believe that a model effectively leveraging both the knowledge from LLMs and data can achieve broader applicability. We have added this discussion to Conclusion.

---

### Review · Reviewer_RxCJ · 2025-02-20

**Summary Of Contributions:**

In this paper, the authors consider the problem of self-supervised representation learning, in particular, focusing on utilizing the downstream task description and the meta-information of the data to discover features relevant to the task, and then, treat the discovered features as ground-truth labels to perform task-specific representation learning.

Problem formulation: The authors consider an unlabeled tabular dataset with $d$-dimensional input features. Then, a downstream task description in natural language, and the names and short descriptions of each feature are provided.

The model aims to train an encoder that extracts data representations to tackle the downstream task in an unsupervised setting (i.e., no ground truth labels are provided).

Experiment: The proposed approach is evaluated across multiple tabular datasets with various downstream tasks.
- The evaluation uses a total of 22 datasets.

- The baselines are all trained in an unsupervised manner as in the experimental setup, including using Raw data, AutoEncoder, SimSiam, SCARF (an augmentation-invariant embedding approach), etc. Compared to all of these baselines, the paper shows that their proposed approach can outperform them in most cases.

- The implementation uses GPT-3.5 as the LLM backbone for feature discovery.

**Audience:**

Yes

**Claims And Evidence:**

No

**Requested Changes:**

In summary, while I see potential in this paper as publication in TMLR, the current writing falls short in the bar as required by the reviewing criteria. Specifically, the following areas can be improved to move towards that:

* Problem formulation is unclear: The authors need to work out a more precise problem definition. Then, the proposed approach should be laid out in more precise language.

After that, within the problem setup, the authors need to clearly describe **under what conditions are the proposed approach expected to work well or not.**

 Without that it's difficult to evaluate the scope and the accuracy of the claims made by this paper.

* Writing lacks clarity: While the paper is mostly carefully written, there are plenty of room for improvement, and there are gaps in the logic of the writing. In particular, I think the authors need to put in effort to improve the quality and the presentation of their work.

Here're some examples (and there're more in the paper):

- In the abstract, the authors introduce TST-LLM, without giving the full name of this acronym. In particular, it's not clear what TST stands for.

- Similarly, the authors refer to the baselines including STUNT and LFT, but did not describe what they are in the abstract.

- On page 3, the authors discuss "in-context learning-based prompt engineering" --- I find this phrase to be particularly confusing. In-context learning usually does not include prompt engineering --- It might also involve prompt selection, but the description here lacks precision.

In summary, I think the paper requires proofreading and appears to be fairly rough in writing.

**Strengths And Weaknesses:**

Strengths:

- I believe this paper is tackling an important problem (identifying features from unlabeled data based on a text description), and utilizing recent LLMs to address this problem sounds like a natural solution.

- The experimental evaluation is solid, mostly positive, and comes with a detailed ablation study. I have not worked on any of the baselines so I could not verify the details, but from a quick glance I have found the analysis to be rigorously conducted.

Weaknesses:

- The proposed approach appears to be ad-hoc, and there is no rigorous justification as to why or how it works. In more detail, the overall approach is presented as an illustrative pipeline figure (Figure 1), and there is no algorithm box, no clear description of what's the input and what's the output for this problem.

This ad-hoc nature limits the generalizability of the proposed approach.

- While the improvement in the experiments appears to be convincing, I do not get a good sense of why this improves over the baselines (such as SimSiam and other SSL methods).

This limits the current contribution of the paper mostly at a data-engineering level, lacking a more solid treatment.

---

> ### Author Response · Authors · 2025-03-03
> **Response to Reviewer RxCJ**
>
> Thank you so much for your valuable feedback. We have carefully revised our paper based on your comments. If you have any further questions, please let us know!
>
> > 1. Problem formulation is unclear. There is no algorithm box, no clear description of what's the input and what's the output for this problem.
>
> To clarify our problem formulation, we have revised the method section by formally defining the problem, explicitly specifying the input and output, and ensuring a clearer understanding of our approach. We've also included pseudo-code-based algorithms to enhance readability and facilitate comprehension of the proposed method (see Algorithms 1 and 2).
>
> > 2. The proposed approach appears to be ad-hoc, and there is no rigorous justification as to why or how it works. I do not get a good sense of why this improves over the baselines. This ad-hoc nature limits the generalizability of the proposed approach.
>
> First and foremost, we want to clarify that our method is a standalone approach for representation learning, rather than an ad-hoc technique. Traditional representation learning methods often define pretext tasks to train models, but these tasks frequently overlook the context of the target problem and the nature of the data, resulting in mismatches.
> To address this issue, our approach utilizes meta-information from each task and refines it using LLM’s prior knowledge to identify task-specific target features through feature engineering. By integrating these refined features into representation learning, our method ensures that the learned representations are better aligned with the task context, ultimately leading to improved performance in downstream tasks.
> To prevent any confusion, we have included an additional description of our method, accompanied by an illustration (see Figure 1). Furthermore, we've provided an example in the introduction section, demonstrating when and why our model performs well, using a diabetes classification task as a case study.
>
> > 3. Writing lacks clarity
>
> Thank you for your valuable feedback in improving our paper’s clarity. Based on your suggestions, we have made several modifications to enhance the readability and comprehensibility of our work. Here are some examples of the changes we made:
>
> * In the abstract, the authors introduce TST-LLM, without giving the full name of this acronym. In particular, it's not clear what TST stands for.
> → We have included the full name of our method (Task-specific Self-supervised Tabular learning with LLMs) in the abstract.
>
>
> * Similarly, the authors refer to the baselines including STUNT and LFT, but did not describe what they are in the abstract.
> → We have removed references to these baselines from the abstract and properly introduced them in the related work and experiment sections.
>
> * On page 3, the authors discuss "in-context learning-based prompt engineering"
> → We have revised the sentence to "in-context learning while calibrating the prompts" to clarify that these works explore better prompts for in-context learning.

---

### Comment · Action_Editor_6BE9 · 2025-02-20
**Discussion Period Open**

Hi Everyone,

The discussion period is now open. Please take this time to discuss the paper amongst each other -- the reviewers will be able to start making recommendations in two weeks.

Thanks,
AE

---

### Decision · Action_Editor_6BE9 · 2025-04-02

**Recommendation:** Accept with minor revision

**Comment:**

There was a majority opinion leaning accept on this paper, and I tend to agree with those reviewers. This work is interesting and presents a well-motivated and reasonable approach to solve the problem of SSL on tabular data. The experimental analysis has become more complete with the help of constructive criticism from the reviewers, and the writing in the paper is now of sufficient quality for TMLR.

I am accepting the paper with the following revisions:
1. Please bring the results of Appendix F.2 up to the main paper.
2. Please provide better justification for the choice of your tabular benchmarks in the main paper.

As for certifications, no reviewers suggested any.

**Audience:**

Yes, the somewhat automatic analysis of tabular data with the help of LLMs remains a hot topic in the tabular field, even if LLM-based tabular processing is still somewhat limited in scope.

**Claims And Evidence:**

There was some discussion about whether the experimental setup was sufficient to prove the merits of this paper. In particular, multiple reviewers asked for comparisons to tree-based techniques (standard in tabular data). One reviewer further noted that the previous baselines did not have the same level of access to the training targets as the proposed method, calling into question the fairness of the comparison; that same reviewer also raised the issue that the benchmarks considered were nonstandard, furthering a typical problem in the tabular data community where people pick and choose benchmarks as they please with no solid reasoning behind the choice.

The authors made a strong response to the concern about the fairness of the comparison by comparing against tree-based methods in the appendix, and added more OpenML benchmarks to the appendix as well. Yet the choice of benchmarks is still somewhat arbitrary and could use further justification, and the results from Appendix F.2 are sufficiently interesting to merit inclusion in the main paper.

---

> ### Author Response · Authors · 2025-04-18
> **Thank you for thoughtful reviews and recognizing the contributions of our work.**
>
> We sincerely appreciate the reviewers and the editor for recognizing the contributions of our work and for their thoughtful feedback, which has been instrumental in improving the quality of our paper. Below is a summary of the revisions we made for the camera-ready version.
>
> > 1. Please bring the results of Appendix F.2 up to the main paper.
>
> As suggested, we have moved the contents of Appendix F.2 to Section 4.2, titled “Comparison with Conventional Baselines,” to facilitate a more accessible and organized comparison with other baselines.
>
> > 2. Please provide better justification for the choice of your tabular benchmarks in the main paper.
>
> We have enhanced the “Datasets” paragraph in Section 4.1 (“Performance Evaluation”) by adding a justification for the selection of our benchmark datasets and clarifying the literature references we consulted. We also included a pointer to Appendix F.1 for results on additional datasets.